# Methylene blue accelerates liquid-to-gel transition of tau condensates impacting tau function and pathology

Yongqi Huang [1,9] ✉, Jitao Wen[2,3,9], Lisa-Marie Ramirez [4],
Eymen Gümüşdil [4,5], Pravin Pokhrel[6], Viet H. Man[7], Haiqiong Ye [1], Yue Han [1],
Yunfei Liu [1], Ping Li [1], Zhengding Su [1], Junmei Wang [7], Hanbin Mao [6],
Markus Zweckstetter [4,8], Sarah Perrett [2,3], Si Wu [2,3] ✉ & Meng Gao [1] ✉

Preventing tau aggregation is a potential therapeutic strategy in Alzheimer's disease and other tauopathies. Recently, liquid–liquid phase separation has been found to facilitate the formation of pathogenic tau conformations and fibrillar aggregates, although many aspects of the conformational transitions of tau during the phase transition process remain unknown. Here, we demonstrate that the tau aggregation inhibitor methylene blue promotes tau liquid–liquid phase separation and accelerates the liquid-to-gel transition of tau droplets independent of the redox activity of methylene blue. We further show that methylene blue inhibits the conversion of tau droplets into fibrils and reduces the cytotoxicity of tau aggregates. Although gelation slows down the mobility of tau and tubulin, it does not impair microtubule assembly within tau droplets. These findings suggest that methylene blue inhibits tau amyloid fibrillization and accelerates tau droplet gelation via distinct mechanisms, thus providing insights into the activity of tau aggregation inhibitors in the context of phase transition.

Tau is a microtubule-associated protein encoded by the *MAPT* gene. Under normal conditions, tau regulates the assembly of microtubules[1,2] and the activity of motor proteins[3,4]. However, in Alzheimer's disease and several other neurodegenerative disorders, tau protein abnormally undergoes manifold posttranslational modifications and its normal function is disrupted, leading to the formation of insoluble fibrils and the development of neurofibrillary tangles in the neuronal soma[1,2,5–7]. The structures of tau filaments from several tauopathies have been determined recently by cryo-electron microscopy, showing distinct folding patterns in different tauopathies[8,9].

Preventing tau aggregation and promoting clearance of tau aggregates represent potential strategies for tau-targeting therapies[10,11]. Tau-binding molecules could block propagation of tau fibrils, trap tau in aggregation-incompetent conformations, and/or

[1]Cooperative Innovation Center of Industrial Fermentation (Ministry of Education & Hubei Province), Key Laboratory of Industrial Fermentation (Ministry of Education), Hubei Key Laboratory of Industrial Microbiology, Hubei University of Technology, 430068 Wuhan, China. [2]National Laboratory of Biomacromolecules, CAS Center for Excellence in Biomacromolecules, Institute of Biophysics, Chinese Academy of Sciences, 100101 Beijing, China. [3]University of the Chinese Academy of Sciences, 100049 Beijing, China. [4]German Center for Neurodegenerative Diseases (DZNE), Von-Siebold-Str. 3a, 37075 Göttingen, Germany. [5]Department of Molecular Biology and Genetics, Gebze Technical University, 41400 Gebze Çayirova, Kocaeli, Turkey. [6]Department of Chemistry & Biochemistry, Advanced Materials and Liquid Crystal Institute, Department of Biomedical Sciences, Kent State University, Kent, OH 44242, USA. [7]Department of Pharmaceutical Sciences and Computational Chemical Genomics Screening Center, School of Pharmacy, University of Pittsburgh, Pittsburgh, PA 15261, USA. [8]Department for NMR-based Structural Biology, Max Planck Institute for Multidisciplinary Sciences, Am Fassberg 11, 37077 Göttingen, Germany. [9]These authors contributed equally: Yongqi Huang, Jitao Wen. ✉e-mail: yqhuang@hbut.edu.cn; wusi@ibp.ac.cn; gaomeng@hbut.edu.cn

dissemble tau fibrils. To date, various molecules have been tested and a number of molecules have been identified that inhibit tau aggregation in vitro and in cells, including methylene blue (MB) and its reduced derivative leuco-methylthioninium bis(hydromethanesulphonate) (LMTM)[12–14] (Fig. 1a). MB and LMTM have been suggested to mainly bind to the microtubule-binding domain of tau, involving Cys291 and Cys322[15,16]. The efficacy of LMTM has been tested in patients suffering from Alzheimer's disease but the compound failed in phase III clinical trials[17–20], highlighting that further insights into tau aggregation mechanisms are required.

Liquid–liquid phase separation (LLPS) of proteins and nucleic acids and the formation of membrane-less organelles have been connected to multiple biological processes and are potentially involved in human diseases[21–25]. In 2017, three studies independently demonstrated that tau readily undergoes LLPS in vitro and forms liquid-like droplets[26–28]. Phase separation of tau causes the crowding of aggregation-prone elements of tau and promotes tau fibrillization[26,29]. The accompanying conformational expansion of tau upon LLPS may also contribute to the oligomerization and aggregation of tau[26,30–35]. In addition, phase-separated tau was found to nucleate microtubule

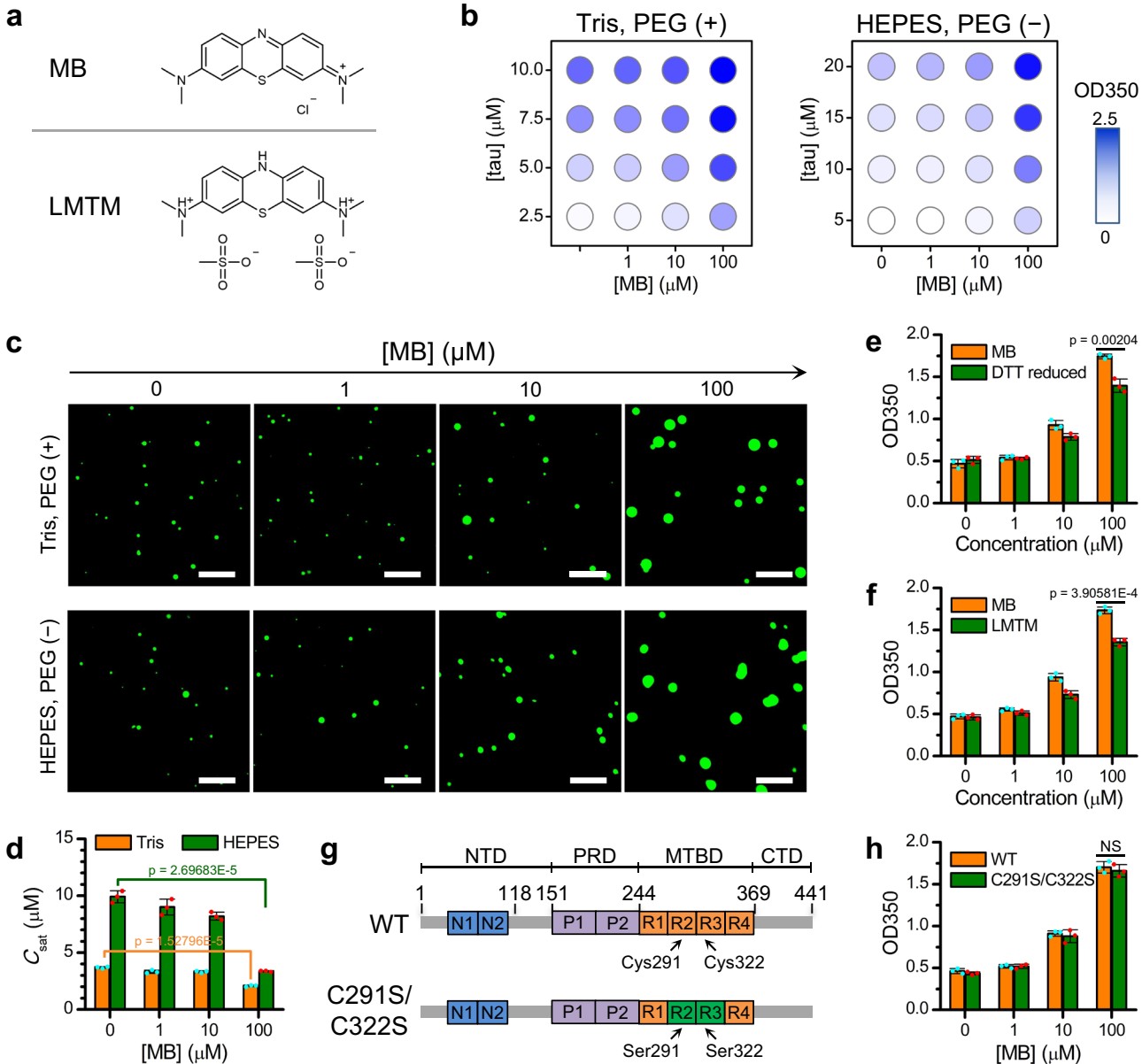

**Fig. 1 | MB facilitates tau phase separation. a** Structure of MB and LMTM. **b** Turbidity of tau solution measured at 350 nm (OD350) for different tau and MB concentrations, as indicated, in 50 mM Tris (pH 7.4) with 5% PEG8000 or 20 mM HEPES (pH 7.4) without PEG8000. **c** Representative fluorescence microscopy images of tau droplets with different MB concentrations. The concentrations of tau were 5 µM in 50 mM Tris (pH 7.4) with 5% PEG8000 and 20 µM in 20 mM HEPES (pH 7.4) without PEG8000. Scale bars, 10 µm. **d** Saturation concentration of tau ($C_{sat}$) with different MB concentrations, as indicated, in 50 mM Tris (pH 7.4) with 5% PEG8000 or in 20 mM HEPES (pH 7.4) without PEG8000. **e** Turbidity of tau solution (5 µM) with different concentrations of MB or DTT-reduced MB, as indicated, in 50 mM Tris (pH 7.4) with 5% PEG8000. **f** Turbidity of tau solution (5 µM) in the presence of a series of concentrations of MB or LMTM, as indicated, in 50 mM Tris (pH 7.4) with 5% PEG8000. **g** Design of the tau cysteine-less mutant. Locations of the two N-terminal inserts (N1 and N2) within the N-terminal domain (NTD), the two proline-rich regions (P1 and P2) within the proline-rich domain (PRD), the four microtubule-binding repeats (R1 to R4) within the microtubule-binding domain (MTBD), and the C-terminal domain (CTD) are indicated. **h** Turbidity of tau solution (5 µM) in the presence of different MB concentrations, as indicated, in 50 mM Tris (pH 7.4) with 5% PEG8000. Data in (**d–f**, **h**) are presented as mean values +/− SD of three experiments. Significance levels were determined by unpaired two-sided Student's *t* test. NS non-significant. Source data are provided as a Source data file.

bundles[28,36] and regulate microtubule functions[37–39]. Thus, tau LLPS has been suggested to be related to both the normal function of tau and the development of neurodegenerative diseases[40–43].

Given the intricate connection between LLPS and fibril formation and the biological implications of LLPS, it is important to investigate the potential influence of aggregation inhibitors on tau LLPS and the activity of inhibitors against tau aggregation in the context of LLPS. Here we focus on the influence of the tau aggregation inhibitor MB on tau phase transition. We find that MB promotes tau LLPS and accelerates the liquid-to-gel transition of tau droplets. These activities of MB are independent of its redox activity and the presence of cysteine residues in tau. While conformational opening of tau is observed in the droplets, the gelated tau droplets do not convert into long fibrils in the presence of MB. We further show that MB reduces the cytotoxicity of tau aggregates in the context of LLPS and gelation of tau droplets induced by MB has little effect on microtubule assembly. Collectively, our results suggest that MB inhibits tau aggregation and accelerates tau droplet gelation via distinct mechanisms. This provides insights into the activity of tau aggregation inhibitors in the context of phase transition.

## Results

### MB promotes tau phase separation
To investigate the potential influence of MB on the phase separation of tau, turbidity of tau solution was measured after incubation with different concentrations of MB in 50 mM Tris supplemented with 5% PEG8000. Our results showed that the turbidity of tau solution increased with MB concentration (Fig. 1b). Fluorescence microscopy was performed using unlabeled tau doped with GFP-fused tau, confirming that the increase in solution turbidity was due to enhanced phase separation (Fig. 1c). To assess the effect of MB on the phase separation propensity of tau, we determined the saturation concentration of tau ($C_{sat}$) under different MB concentrations using centrifugation[44]. As shown in Fig. 1d, $C_{sat}$ decreased with MB concentration, suggesting that MB enhances the capacity of tau to undergo phase separation.

PEG is a molecular crowding agent that facilitates phase separation. To exclude the possibility that the observed activity of MB on tau phase separation was due to the presence of crowding agents, we carried out the above experiments in 20 mM HEPES where tau undergoes LLPS without PEG. Consistent with results in Tris buffer, the solution turbidity and the size of droplets increased as the concentration of MB was increased in HEPES buffer (Fig. 1b, c). Similarly, $C_{sat}$ decreased with MB concentration in HEPES buffer (Fig. 1d).

Previous studies show that the major binding sites of MB on tau are adjacent to the two cysteine residues (i.e., Cys291 and Cys322) within the microtubule-binding repeats R2 and R3[15,16]. Upon binding to tau, MB oxidizes cysteine residues, which has been proposed to be vital for the inhibitory activity of MB against full-length tau aggregation[15,45,46], although MB inhibition of the aggregation of tau dGAE fragment (Ile297–Glu391) could be cysteine-independent[47]. To determine whether MB enhances tau phase separation by oxidizing the cysteine residues, we reduced MB with DTT (Supplementary Fig. 1a, b). We confirmed that the DTT-reduced MB promotes phase separation of tau (Fig. 1e, and Supplementary Fig. 1c). Increasing the concentration of the reduced MB derivative LMTM also increased the turbidity of tau solution (Fig. 1f). However, there was a difference in solution turbidity between MB and reduced MB (Fig. 1e, f). We further generated a cysteine-less tau mutant (tau$_{C291S/C322S}$) by replacing the two cysteine residues with serine (Fig. 1g). Consistent with previous work[48], cysteine-to-serine replacement does not affect the LLPS propensity of tau (Supplementary Fig. 2a). Similar to wild-type (WT) tau, the turbidity of tau$_{C291S/C322S}$ solution increased upon incubating with increasing concentrations of MB in both Tris and HEPES buffers (Fig. 1h and Supplementary Fig. 2b). Fluorescence microscopy images further

confirmed that droplets were formed by tau$_{C291S/C322S}$ and their sizes were similar to those of WT tau (Supplementary Fig. 2c).

We noted that there was no significant difference in solution turbidity between MB/tau$_{C291S/C322S}$ and MB/WT tau (Fig. 1h), or between reduced MB/tau$_{C291S/C322S}$ and reduced MB/WT tau (Supplementary Fig. 1d). Therefore, the difference in solution turbidity between MB/WT tau and reduced MB/WT tau (Fig. 1e, f) could result from the change in interactions between MB and tau upon reduction of MB. Collectively, our results show that MB enhances the formation of tau droplets, with the redox activity of MB or the cysteine residues of tau contributing only a minor role.

### MB accelerates the liquid-to-gel transition of tau droplets
Next, we determined whether MB has an influence on the fluidity of tau droplets by performing fluorescence recovery after photobleaching (FRAP) experiments. Consistent with previous reports[31,48,49], the bleached region within the tau droplet recovered rapidly in the absence of MB (Fig. 2a, b and Supplementary Fig. 3). Unexpectedly, the recovery of tau droplets in the presence of MB was reduced. Recovery of the bleached region was not observed when the MB concentration was increased to 10 μM in HEPES buffer (Fig. 2a, b). Although the extent of recovery was larger in Tris buffer, almost no recovery was observed when the MB concentration was increased to 100 μM (Supplementary Fig. 3). We also performed FRAP experiments for tau$_{C291S/C322S}$ droplets. Similar to WT tau, the recovery of tau$_{C291S/C322S}$ droplets was reduced when the concentration of MB was increased (Supplementary Fig. 4). Therefore, the mobility of tau molecules inside the droplets was decreased when MB was present.

The above FRAP results suggest that MB promotes the liquid-to-gel transition of tau droplets. To further confirm the MB induced gelation of tau droplets, we measured the aspect ratio of fusing droplets using optical tweezers[50] at various time points at different MB concentrations (Fig. 2c, d). In the absence of MB, tau droplets incubated for about 10 min completely fused as the aspect ratio of the fused droplet was close to 1.0, which implies that the tau droplets are in a liquid-like state. The aspect ratio increased to 1.6 when the incubation time was increased to 180 min, indicating that a liquid-to-gel transition occurred during the aging process. Increasing the MB concentration decreased the extent of fusion between two droplets and the aspect ratio increased. In the presence of 10 μM MB, an ellipsoidal droplet with an aspect ratio of 1.6 was observed after 10 min incubation. The fused droplets became dumbbell shaped and the aspect ratio was larger than 1.9 when the incubation time was increased to 180 min. Fitting the aspect ratio over incubation time enabled us to estimate the halftime of the liquid-to-gel transition. It was found that increasing the concentration of MB reduced the halftime from 94 to 6 min (Supplementary Table 1).

MB has several derivatives, such as azure A and azure B (Fig. 3a), which also inhibit tau aggregation[12,51]. To explore whether these two MB derivatives can regulate tau phase separation, we measured the turbidity of tau solutions in the presence of a series of concentrations of azure A or azure B. Similar to MB, incubation of WT tau or tau$_{C291S/C322S}$ with azure A or azure B increased the solution turbidity in a concentration-dependent manner (Fig. 3b and Supplementary Figs. 5 and 6). At a concentration of 100 μM, the effect of azure A on solution turbidity was comparable to MB, whereas the effect of azure B was weaker than MB. Fluorescence microscopy images confirmed that larger droplets formed when the concentrations of azure A and azure B were increased (Supplementary Fig. 7). FRAP experiments showed that azure B has the strongest efficacy in inducing gelation of tau droplets (Fig. 3c and Supplementary Fig. 8). Saturation concentration measurements showed that azure A has similar effects as MB on reducing $C_{sat}$ (Fig. 3d). In contrast, $C_{sat}$ only changed marginally as the concentration of azure B was increased, indicating that

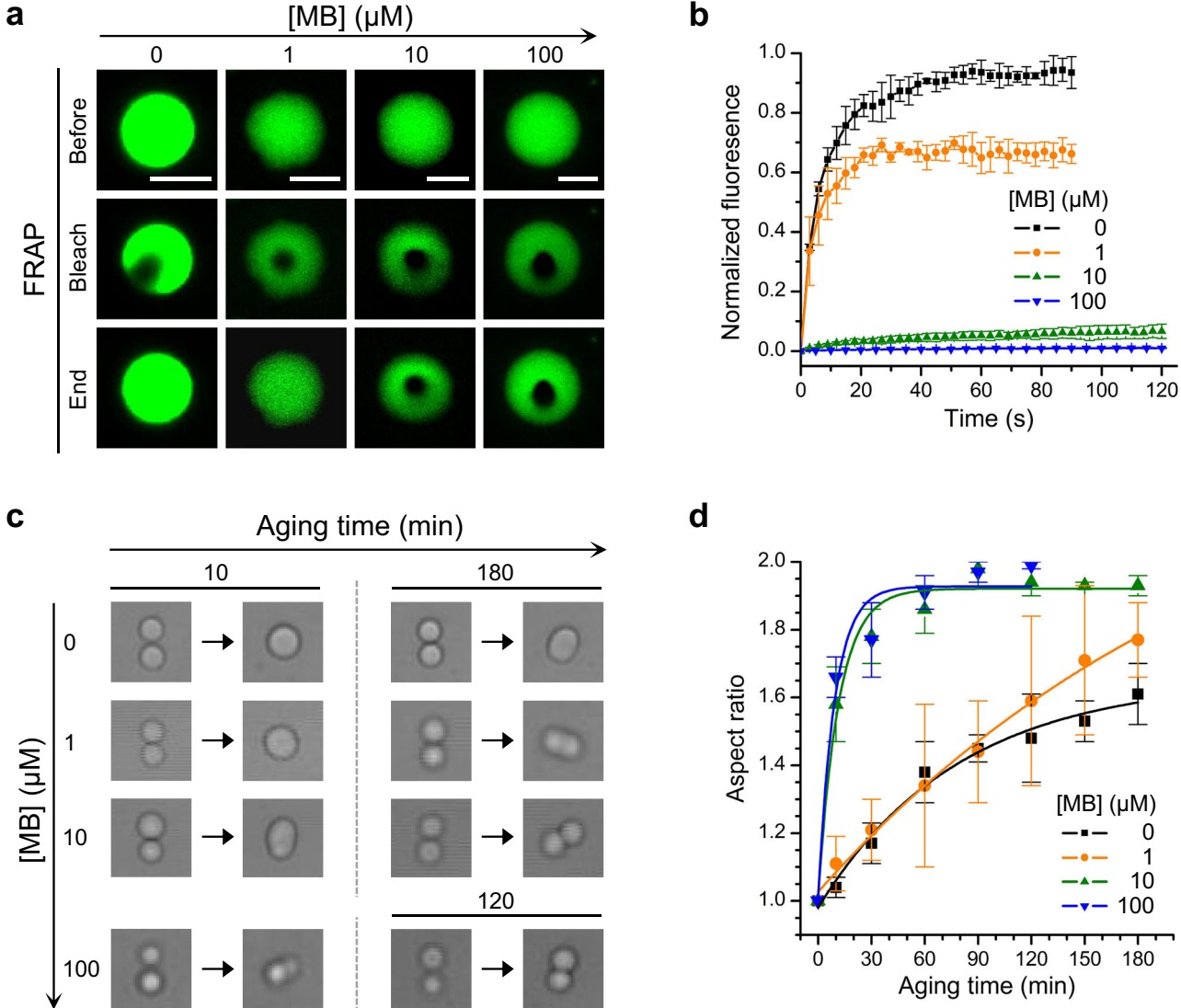

**Fig. 2 | MB facilitates liquid-to-gel transition of WT tau droplets.**
**a** Representative FRAP images of tau droplets formed by 20 µM tau and different concentrations of MB, as indicated, in 20 mM HEPES (pH 7.4) without PEG8000. Scale bars, 5 µm. **b** Quantified fluorescence intensity of the FRAP experiments. Data are presented as mean values +/− SD of three experiments. **c** Fusions of two tau droplets measured using optical tweezers in 25 mM HEPES (pH 7.4), 150 mM KCl, 1 mM DTT, 15% PEG6000 and different concentrations of MB, as indicated. To obtain droplets of suitable size for fusion kinetics measurements, PEG was added to facilitate tau droplet formation. **d** Aspect ratio of the two joined tau droplets as a function of aging time in the presence of different MB concentrations, as indicated. The lines are single exponential fitting of the data points. Refer to Supplementary Table 1 for further details of data fitting. Data are presented as mean values +/− SD of at least three experiments. Source data are provided as a Source data file.

azure B enhances dynamic arrest of tau droplets with minimal effect on the capacity of tau to undergo phase separation. In contrast to $C_{sat}$, the concentration of tau inside the droplets did not significantly vary for the different compounds (Supplementary Fig. 9). By comparing the concentration of compounds in the dilute phase and the condensed phase (Supplementary Fig. 10), the compounds were found to be enriched in tau droplets. We recently showed that liquid-to-gel transition of tau droplets can be significantly attenuated by trimetlylamine oxide (TMAO) likely due to direct interactions between TMAO and tau[50]. However, Fig. 3e showed that TMAO, at a concentration up to 0.2 M, could not mitigate the liquid-to-gel transition of tau droplets in the presence of MB, azure A or azure B.

Altogether, these experiments clearly demonstrate that MB promotes the liquid-to-gel transition of tau droplets and reduces the mobility of tau molecules inside the droplets in a concentration-dependent manner. Consistent results obtained from different experimental conditions indicate that the regulatory activity of MB on tau LLPS is an intrinsic property of the compound.

## MB binds to broad regions of tau through hydrophobic and electrostatic interactions

Although the two intrinsic cysteine residues have been suggested to play a key role in inhibition of tau aggregation by MB[15,45,46], our above results demonstrate that they are not essential for MB to promote LLPS of tau and the liquid-to-gel transition of tau droplets. To dissect the underlying interactions between MB and tau, we first treated phase-separated tau solutions with NaCl or 1,6-hexanediol (1,6-HD). In the absence of MB, a strong decrease in phase separation was observed when the tau solution was treated with NaCl (Fig. 4a, left panel, green, and Supplementary Fig. 11a), whereas 1,6-HD treatment resulted in a slight decrease in phase separation (Fig. 4a, right panel, green bars, and Supplementary Fig. 11b). In contrast, in the presence of 100 µM MB,

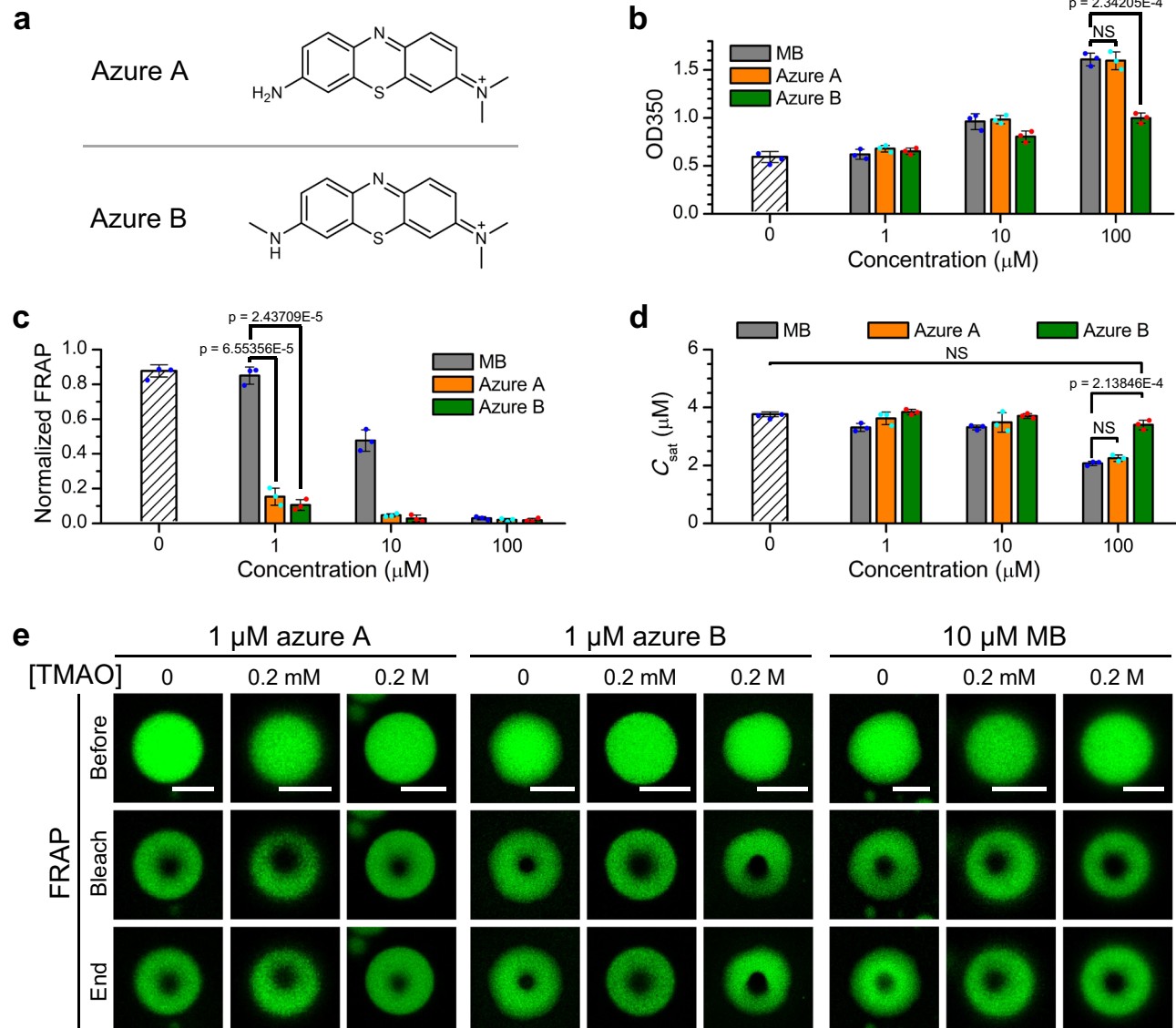

**Fig. 3 | Influence of MB derivatives on WT tau phase separation. a** Structures of MB derivatives azure A and azure B. **b** Turbidity of 5 µM tau solution with different concentrations of MB, azure A, and azure B, as indicated, in 50 mM Tris (pH 7.4) with 5% PEG8000. **c** Normalized FRAP intensity at the end of FRAP experiments. Droplets were formed by 10 µM tau in 50 mM Tris (pH 7.4) with 5% PEG8000 and different concentrations of MB, azure A, and azure B. **d** Saturation concentration of tau in the presence of different concentrations of MB, azure A, or azure B in 50 mM Tris (pH 7.4) with 5% PEG8000. **e** FRAP experiments for tau droplets with different concentrations of TMAO, as indicated, in the presence of 1 µM azure A, 1 µM azure B, or 10 µM MB. Tau droplets were formed by 10 µM tau in 50 mM Tris (pH 7.4) with 5% PEG8000. Scale bars, 5 µm. Data in **b, c, d** are presented as mean values +/− SD of three experiments. Significance levels were determined by unpaired two-sided Student's *t* test. NS non-significant. Source data are provided as a Source data file.

treatment of tau solution with NaCl or 1,6-HD both resulted in a significant decrease in phase separation (Fig. 4a, orange bars, and Supplementary Fig. 11). Since NaCl screens electrostatic interactions and 1,6-HD is thought to disrupt hydrophobic interactions[52–55], the sensitivity of MB-induced tau droplets to NaCl and 1,6-HD suggests that MB promotes tau phase separation via both electrostatic interactions and hydrophobic interactions.

To identify the binding regions of MB on tau, we constructed a number of tau deletion variants (Fig. 4b). Deletion of any of these domains (ΔNTD, ΔMTBD, ΔPRD, or ΔCTD) reduced the propensity of tau to undergo LLPS (Fig. 4c and Supplementary Fig. 12, 0 µM MB), in agreement with previous data[26,39,49,56]. Importantly, the extent of LLPS for all four tau deletion variants increased as the MB concentration was increased (Fig. 4c and Supplementary Fig. 12) and the droplets underwent liquid-to-gel transition in the presence of MB (Supplementary Fig. 13). Notably, the turbidity increase for tau ΔNTD was

smaller than for WT tau and the other deletion variants, which may result from a dramatic loss of negative charges that are essential for tau LLPS.

$^1$H-$^{15}$N SOFAST HMQC spectra[57] of $^{15}$N-labeled WT tau protein were recorded in the absence and presence of MB (Fig. 4d and Supplementary Fig. 14b). The NMR spectra were recorded at 5 °C to minimize loss of $^1$H signal due to solvent exchange[58]. Under these conditions, tau did not form droplets in the absence of MB (Fig. 4e). Addition of MB resulted in the formation of a biphasic sample consisting of suspended tau droplets in a dispersed phase of tau (Fig. 4e). Quantification of the changes in peak intensities upon MB addition revealed an overall reduction in NMR signal intensities involving all domains of tau (Fig. 4f and Supplementary Fig. 14f). A reduction in NMR signal intensity in relation to protein phase separation might arise from the slower tumbling of the protein in the droplet phase and associated shortened transverse relaxation times[59]. In addition, exchange between an NMR-

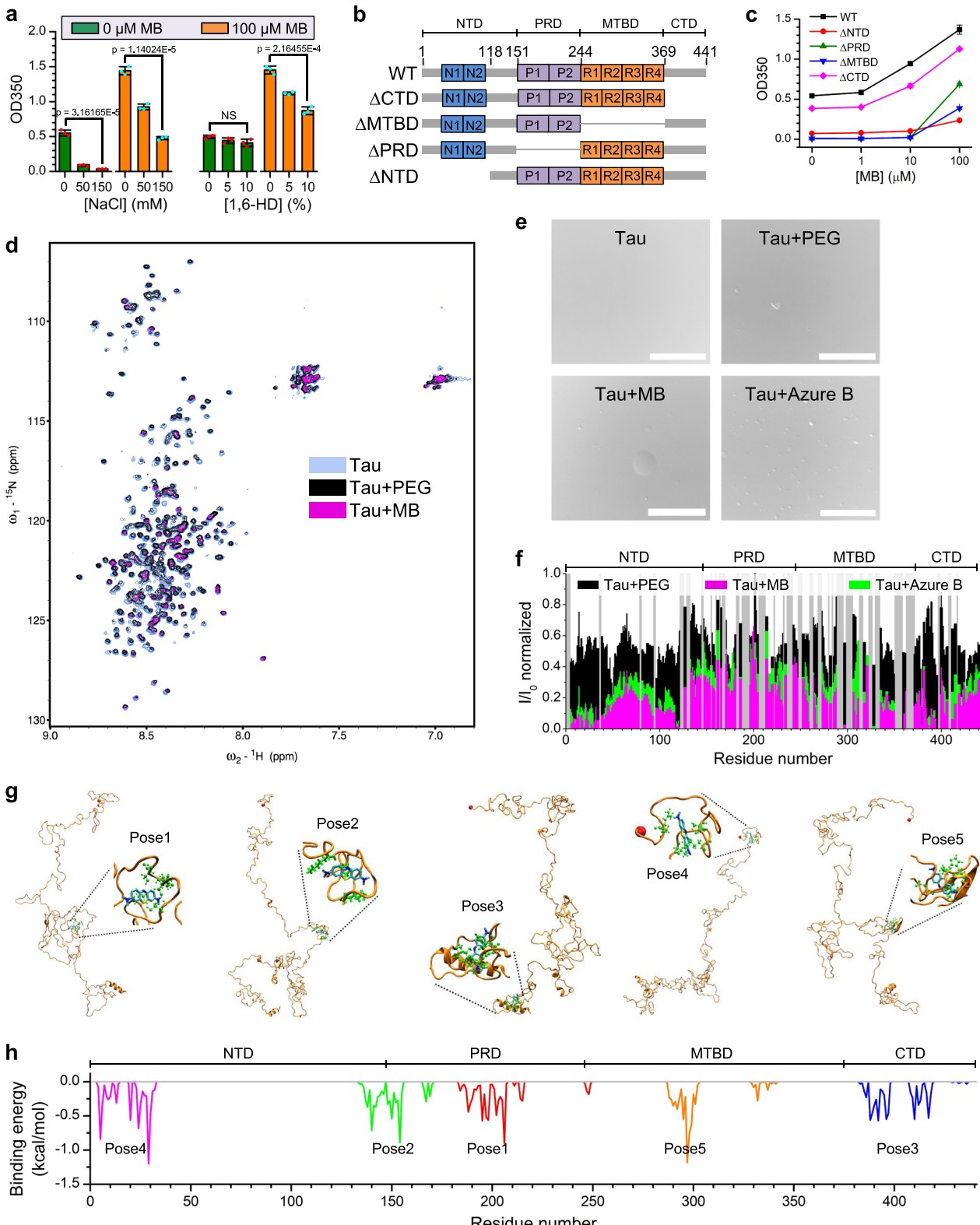

visible state of the protein (i.e., monomeric tau in the dispersed phase) and an NMR-invisible state (i.e., droplets or aggregates) potentially contributes to the reduction in signal. In the case of protein-ligand binding, NMR line broadening may also occur due to exchange between free and ligand-bound states of the protein in the intermediate exchange regime. To assess the impact of droplet formation without MB binding on the NMR spectrum of tau, we collected the $^{1}$H-$^{15}$N SOFAST HMQC spectrum of tau in the presence of 5% PEG8000

(Fig. 4d and Supplementary Fig. 14a). PEG induced tau droplet formation (Fig. 4e) and led to ~40% peak amplitude reduction (Fig. 4f and Supplementary Fig. 14d) as well as chemical shift perturbations (Supplementary Fig. 14e) across all tau domains, consistent with the formation of a slow-tumbling NMR-invisible state (i.e., droplet phase). Notably, the intensity ratio profile for the tau-PEG mixture does not completely correspond to that of the tau-MB sample (Fig. 4f). The prominent lack of overlay for the entire NTD and CTD regions suggest

**Fig. 4 | Interactions between tau and MB. a** Turbidity of WT tau (5 μM) measured at different NaCl or 1,6-HD concentrations in 50 mM Tris (pH 7.4) with 5% PEG8000. The concentrations of MB were 0 and 100 μM. Data are presented as mean values +/− SD of three experiments. Significance levels were determined by unpaired two-sided Student's *t*-test. NS non-significant. **b** Design of tau deletion mutants. **c** Turbidity of tau deletion variants (5 μM) measured at different MB concentrations in 50 mM Tris (pH 7.4) with 5% PEG8000. Data are presented as mean values +/− SD of three experiments. **d** Overlay of 2D $^1$H-$^{15}$N SOFAST-HMQC spectra of WT tau (18 μM) with and without 5% PEG8000 or 360 μM MB in 50 mM Tris (pH 7.4); ppm

parts per million. **e** Differential interference contrast (DIC) micrographs of the NMR samples. Scale bars, 20 μm. **f** Peak amplitude ratios ($I/I_0$) quantified for tau with 5% PEG8000, 360 μM MB, or 360 μM azure B calculated from the $^1$H-$^{15}$N SOFAST-HMQC spectra. Residues omitted from the $I/I_0$ analysis due to lack of reliable peak assignment are designated as gray bars. **g** The top five docking poses of MB binding to WT tau. The N-terminus of tau is shown as a red ball. The residues which have strong interaction with MB are shown as green balls and sticks. **h** Tau residue-ligand binding profile calculated using the top five docking poses for MB. Source data are provided as a Source Data file.

that these domains are the most affected by MB binding. A patch of the MTBD (residues ~330–350) and some residues in the PRD also appear to bind to MB. We also collected the $^1$H-$^{15}$N SOFAST HMQC spectrum of tau in the presence of azure B (Fig. 4d and Supplementary Fig. 14c). Comparing the intensity ratio profiles for tau-MB and tau-azure B (Fig. 4f), we observed that the profiles are mostly consistent, and that $I/I_0$ attenuation due to MB is more pronounced than that due to azure B except for short stretches of residues such as those around ~200 in the PRD and ~380–400 in the CTD, suggesting that interaction with these residues may be responsible for the different effects of MB and azure B on inducing tau LLPS and droplet gelation. Addition of PEG8000 together with MB or azure B resulted in severe attenuation of the NMR peak intensities due to enhancement of phase separation (Supplementary Fig. 14g, h), which prohibited further residue-level analysis.

To obtain atomic models for the tau-MB complex state, docking simulations were performed. Considering that MB may have a few dominant binding modes with tau, we focused on the top 5 docking poses (Fig. 4g) and calculated the contribution of each residue to ligand binding through docking score decomposition analysis. Consistent with NMR characterization and turbidity analysis, docking results showed that MB interacts with various regions of tau, including the two termini of the NTD, the central region of the PRD, the R2 region of the MTBD, and the N-terminal half of the CTD (Fig. 4h). We further analyzed binding of azure A, azure B, and TMAO with tau using docking simulations (Supplementary Figs. 15 and 16). Similar to MB, the interactions between azure A/azure B and tau were distributed throughout the tau protein. The total binding energies calculated from all 1000 docking poses were −348.78, −367.20, −355.78, and −193.26 kcal/mol for MB, azure A, azure B, and TMAO, respectively, indicating that the interactions between TMAO and tau are much weaker than those for MB, azure A, and azure B.

Taken together, the combined data indicate that the interactions essential for MB to promote LLPS of tau and the liquid-to-gel transition of tau droplets are distributed throughout the whole protein rather than being restricted within particular regions of tau.

### Conformational changes of tau upon MB binding and phase separation

To reveal the underlying mechanism by which MB regulates the phase transition of tau, we used fluorescence resonance energy transfer (FRET) to detect the potential conformational changes in tau upon MB binding. We constructed four tau variants by introducing cysteine residue pairs in T17C/Q244C, T149C/Q244C, Q244C/I354C and I354C/S433C mutations (referred to as tau$_{17/244}$, tau$_{149/244}$, tau$_{244/354}$ and tau$_{354/433}$) while mutating the two intrinsic cysteine residues Cys291 and Cys322 to serine as has been used in our previous studies[35,60], and then labeled the proteins with the AlexaFluor350 (AF350) and Alexa-Fluor488 (AF488) dye pair (Fig. 5a). The AF350/AF488-labeled tau was enriched in the condensed phase upon LLPS (Supplementary Fig. 17). In Tris buffer, we observed significant FRET signals for all four tau variants in their native state (Fig. 5b, black lines), indicating that AF350 and AF488 were within the distance that can undergo FRET. This is consistent with the paper-clip model of the native tau conformation[61,62]. Incubation of tau with MB in the non-LLPS state

resulted in slight changes in the apparent FRET signals (Fig. 5b, green lines, and Supplementary Fig. 18a), suggesting that MB binding perturbed the conformations of tau. This is consistent with the above results from NMR and docking, which indicate extensive interactions between MB and tau (Fig. 4). Inducing tau phase separation by adding PEG decreased the FRET signal of the labeled tau variants (Fig. 5b, blue lines). A further decrease in the apparent FRET signal was observed when tau was co-incubated with MB and PEG, which led to an obvious LLPS state (Fig. 5b, red lines). Two-photon microscopy images further demonstrated that the FRET decrease occurs within the droplets upon addition of MB (Fig. 5c, d). In HEPES buffer, a similar phenomenon was observed although the crowding reagent PEG8000 was not necessary for LLPS. The FRET signals of dual-color labeled tau variants showed a slight decrease in the presence of MB in the non-LLPS state (Fig. 5e, blue lines, and Supplementary Fig. 18b). A marked decrease in the FRET signal was observed when LLPS occurred (Fig. 5e, red lines). To check whether the decrease of the apparent FRET efficiency was caused by artificial quenching of the acceptor (i.e., AF488), we performed fluorescence lifetime measurements, which showed no significant change of AF488 lifetime upon addition of MB in either the native or LLPS states (Supplementary Table 2). Therefore, our FRET analysis indicates that the distance between the labeling pair increases upon MB-induced LLPS especially for the NTD and CTD regions, indicating that the tau conformation changes to an expanded state, which could result from a synergistic effect of phase separation and MB binding.

### MB inhibits conversion of gelated tau droplets into fibrils and reduces tau cytotoxicity

LLPS promotes tau aggregation by crowding of the aggregation-prone repeat region and opening of the tau conformation, where liquid-to-gel transition of tau droplets is potentially on-path to fibril formation[26,29,31,35]. Indeed, several studies have shown that tau fibrils can form during the aging of tau droplets[26,31,32]. Here, we investigated whether MB influences amyloid formation of tau in the context of LLPS. We initiated tau aggregation by heparin and monitored the aggregation kinetics by thioflavin T (ThT) fluorescence. We confirmed that MB facilitates tau droplet formation in the presence of heparin (Supplementary Fig. 19a, b). In the absence of MB, the ThT fluorescence intensity increased rapidly (Fig. 6a, black curves). Increasing the MB concentration from 1 to 100 μM markedly reduced the ThT fluorescence (Fig. 6a). To confirm that fibril formation was inhibited under phase-separating conditions in the presence of 100 μM MB, the morphology of samples incubated for 0, 18, or 36 h was characterized by transmission electron microscopy (TEM) (Fig. 6b). In the absence of MB, long tau fibrils were observed after 18 h incubation in both Tris buffer and HEPES buffer. In contrast, in the presence of MB amorphous aggregates or granular tau oligomers were observed after incubation for 18 and 36 h. Furthermore, we performed ThT fluorescence imaging of the phase-separated tau by confocal microscopy and found that with increasing concentration of MB, the formation of ThT-stainable tau aggregates inside tau droplets was inhibited (Fig. 6c and Supplementary Fig. 20). Therefore, our results show that MB retains its capability to inhibit tau fibril formation, even though MB augments LLPS of tau and accelerates the gelation of tau droplets.

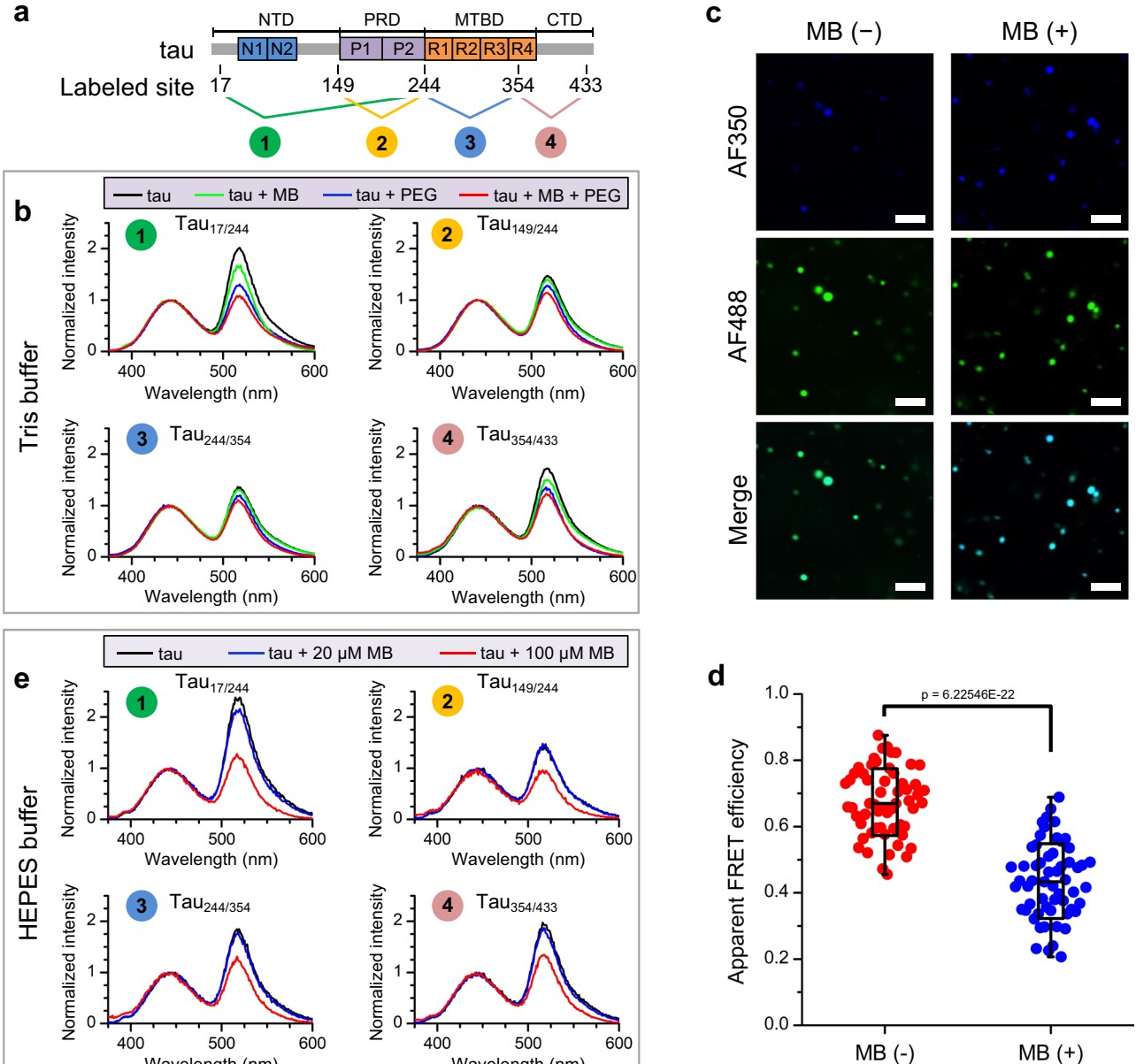

**Fig. 5 | FRET measurements of AF350/AF488-labeled tau variants reveal conformational changes in tau upon MB-induced LLPS. a** Schematic illustration of the fluorescence labeling sites in the four tau variants. **b** FRET spectra of tau variants in 50 mM Tris (pH 7.4). Black: 10 μM tau (no phase separation); green: 10 μM tau + 100 μM MB (no phase separation); blue: 10 μM tau + 5% PEG8000 (phase separation); red: 10 μM tau + 100 μM MB + 5% PEG8000 (strong phase separation). **c** Two-photon microscopy images of AF350/AF488-labeled tau17/244 with or without 100 μM MB in 50 mM Tris (pH 7.4) with 5% PEG8000. Scale bars, 10 μm. **d** Apparent FRET efficiency of tau droplets measured by the two-photon microscopy images under excitation of AF350 by 700 nm laser. Individual data points represent each droplet. A total of 58 droplets were analyzed. The bars indicate minimum and maximum. The center and bounds of box indicate median and SD, respectively. Significance levels were determined by unpaired two-sided Student's *t* test. **e** The FRET spectra of tau variants in 20 mM HEPES (pH 7.4) without PEG8000. Black: 15 μM tau (weak phase separation); blue: 15 μM tau + 20 μM MB (weak phase separation); red: 15 μM tau and 100 μM MB (strong phase separation). A concentration of 0.2 μM AF350/AF488-labeled tau variants was mixed as dopant in all the above solutions for fluorescence spectra measurements and imaging. Source data are provided as a Source data file.

To determine whether MB has the capability to protect cells in the context of LLPS, we measured the cytotoxicity of tau incubated under different conditions over time using the MTT assay. Tau was first incubated with or without MB for different times in the presence of heparin. Human neuroblastoma (SH-SY5Y) cells were then treated with the incubated tau species for 24 h and their viability was assessed. In the absence of MB, at 0 h tau was in the native state with a few small liquid-like droplets, at 5 h tau was in the process of fibril growth, and at 24 h tau was in the fibril state. We took the samples at the above time points along the aggregation reaction and checked the cytotoxicity of tau species. Compared to the control where only buffer including heparin was added into the culture medium, the cell viability slightly decreased upon adding native tau protein and further decreased when incubated with samples taken at 5 h and 24 h along the aggregation time course (Supplementary Fig. 19c). In the presence of MB, where LLPS and gelation occurred, the cytotoxicity of tau species decreased slightly at the same time points as without MB (Supplementary Fig. 19c). These results suggest that MB-induced LLPS of tau, which leads to the formation of gel-like species, alleviates the cytotoxicity caused by tau oligomers and fibrils formed in the absence of MB.

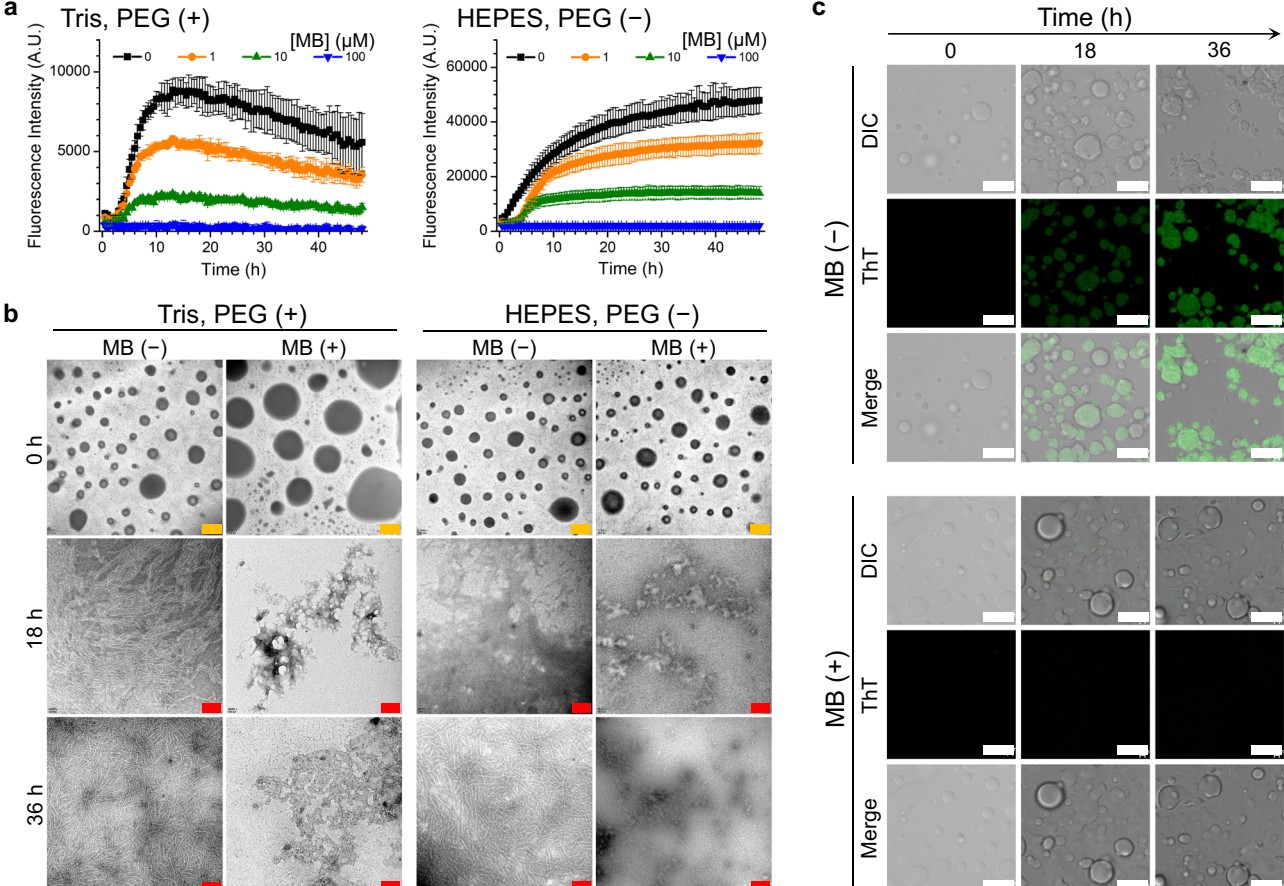

**Fig. 6 | MB inhibits tau fibrillization in the context of phase separation.**
**a** Aggregation kinetics of tau monitored by ThT fluorescence at different MB concentrations in 50 mM Tris (pH 7.4) with 7.5% PEG8000 or in 20 mM HEPES (pH 7.4) without PEG8000. Data are presented as mean values +/− SD of three experiments. A.U., arbitrary units. **b** Representative TEM images of tau aggregates after incubation at 37 °C in 50 mM Tris (pH 7.4) with 7.5% PEG8000 or in 20 mM HEPES

(pH 7.4) without PEG8000, in the absence or presence of 100 μM MB. Scale bars for 0 h represent 1 μm and for 18 and 36 h represent 200 nm. **c** Representative microscopy images of the growth of tau aggregates inside droplets in the absence or presence of 100 μM MB in 50 mM Tris (pH 7.4) with 7.5% PEG8000. Scale bars, 10 μm. Source data are provided as a Source data file.

## Gelation does not impair microtubule assembly within tau droplets

As a major microtubule-binding protein, tau regulates the assembly of tubulin into microtubules. Recent studies provide evidence suggesting that LLPS of tau may be functionally involved in microtubule assembly[28,36–38]. To determine whether liquid-to-gel transition influences the assembly of microtubules within tau droplets, we first mixed tubulin with preformed tau droplets without MB on ice to allow recruitment of tubulin into tau droplets while prohibiting tubulin assembly. We then added MB to a final concentration of 100 μM, or added the same volume of water as a control, and incubated the tau/tubulin droplets on ice for another 15 min. After that, tubulin assembly was triggered by adding GTP and incubating at 37 °C. Fluorescence microscopy images confirmed recruitment of tubulin into tau droplets before tubulin assembly was triggered (Fig. 7a, 0 min). We further confirmed that MB induced gelation of tau droplets by FRAP experiments and found that the mobility of both tau and tubulin was reduced upon MB addition (Fig. 7b, c). Consistent with previous reports[28,36], the morphology of tau droplets changed over time and microtubule bundles appeared upon incubation at 37 °C for 30 min in the absence of MB (Fig. 7a). Importantly, while the mobility of both tau and tubulin was reduced, microtubule bundles also appeared upon incubation at 37 °C for 30 min in the presence of MB and showed no significant difference to those formed in the absence of MB (Fig. 7a). To quantify the influence of MB on the microtubule assembly kinetics, the

assembly process was monitored by the fluorescence of diamidino-phenylindole (DAPI)[63,64], which binds to tubulin and exhibits fluorescence enhancement upon association. The affinity of DAPI for tubulin increases when tubulin assembles into microtubules, thus leading to a further enhancement in DAPI fluorescence. We first confirmed that DAPI has minimal effect on the FRAP recovery of tau droplets and DAPI is co-localized with tubulin inside tau droplets (Fig. 7d–f). As shown in Fig. 7g, the DAPI fluorescence increased rapidly in the first 10 min and reached a plateau phase. The half-time at which the DAPI fluorescence reached 50% of the plateau value increased from 5.6 to 7.9 min when MB was added (Fig. 7g, inset). Therefore, our results show that gelation of tau droplets induced by MB slightly retards the assembly kinetics of tubulin within tau droplets without affecting the final appearance of microtubule bundles.

## Discussion

In this study, we investigated the influence of the aggregation inhibitor MB on the phase transition behavior of tau. Our results demonstrate that MB promotes tau LLPS and decreases the dynamics of tau droplets. Such loss of dynamics of condensates has previously been termed gelation[65–68]. While gelation of liquid droplets has been observed on-path to fibril formation for many proteins[69–73], the gelation of tau droplets facilitated by MB did not result in formation of long ThT-positive amyloid fibrils (Fig. 8a). Although cysteine oxidation induced by MB has been proposed to be vital for the inhibitory activity

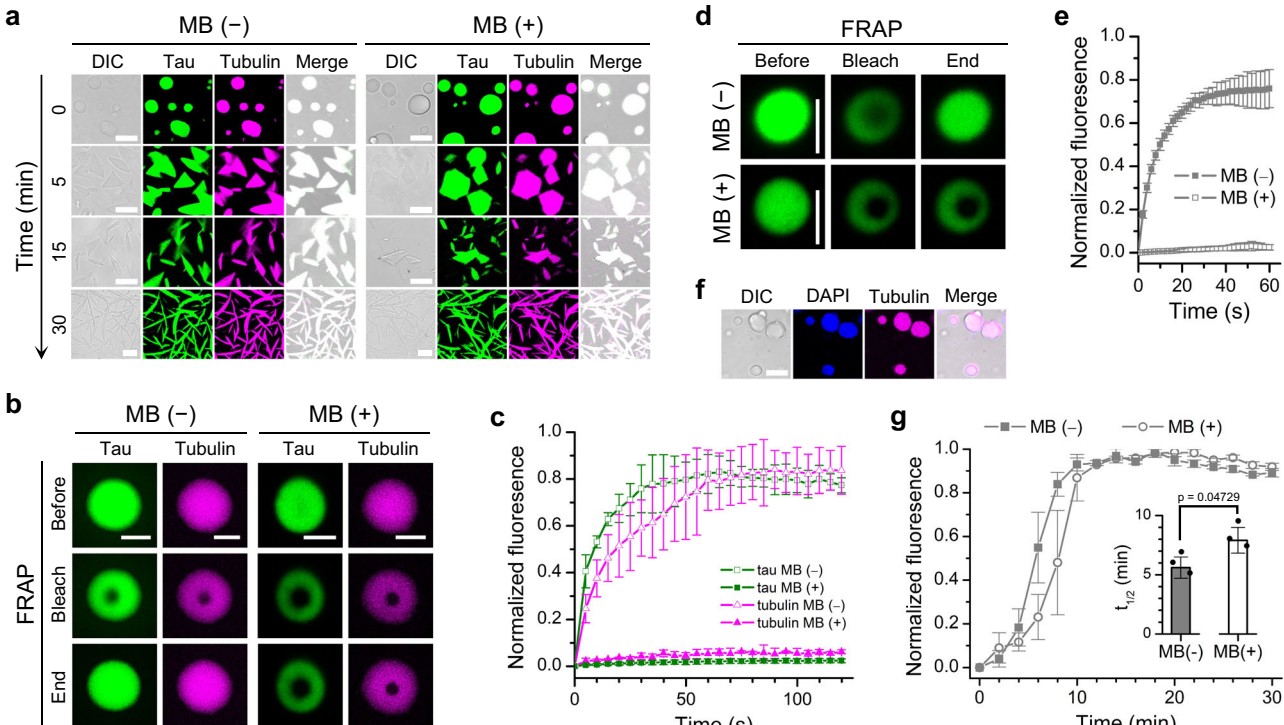

**Fig. 7 | Influence of tau droplet gelation on microtubule assembly.**
**a** Representative microscopy images of the growth of microtubules inside the preformed tau droplets with or without 100 µM MB. Scale bars, 10 µm. **b** Representative FRAP images of tau/tubulin droplets with or without 100 µM MB. Scale bars, 5 µm. **c** Quantified fluorescence intensity of the FRAP experiments of tau/tubulin droplets with or without 100 µM MB. **d** Representative FRAP images of tau/tubulin droplets with or without 100 µM MB in the presence of 1 µM DAPI. Scale bars, 5 µm. **e** Quantified fluorescence intensity of the FRAP experiments of tau/

tubulin droplets with or without 100 µM MB in the presence of 1 µM DAPI. **f** Representative microscopy images showing the co-localization of DAPI and tubulin within tau droplets. Scale bars, 10 µm. **g** Microtubule assembly kinetics with or without 100 µM MB monitored by DAPI fluorescence. Inset: Influence of MB on the half-time of the DAPI fluorescence. Significance levels were determined by unpaired two-sided Student's *t* test. Data in **c**–**g** (inset) are presented as mean values +/− SD of three experiments. Source data are provided as a Source data file.

of MB against tau aggregation[15,45,46], we found that the redox activity of MB and the cysteine residues of tau are not essential for the MB-promoted tau LLPS or the liquid-to-gel transition of tau droplets. The contrasting activities of MB on tau aggregation and LLPS are unexpected and are distinct from previous reports on other ligands[27,48,74–76], suggesting that MB may modulate tau aggregation and phase separation through distinct mechanisms.

To dissect the mechanism by which MB modulates tau phase transition, we constructed various tau deletion variants and applied a variety of biophysical characterization techniques. Turbidity measurements and FRAP experiments for tau deletion variants showed that MB does not regulate tau phase transition via a particular region of tau. Consistent with this, NMR spectroscopy and docking simulations revealed that MB binds to broad regions of tau. Therefore, multiple MB molecules may bind to one tau molecule simultaneously. FRET results further showed that binding of MB to tau disrupts the transient interdomain interactions within tau and induces opening of the paper-clip conformation of tau in the LLPS state[61,62].

We recently found that the liquid-to-gel transition of tau droplets can be driven by the loss of water surrounding tau molecules and TMAO can slow down the gelation of tau droplets[50]. However, TMAO could not attenuate the liquid-to-gel transition of tau droplets when MB was present, suggesting that gelation of tau droplets induced by MB is not due to dehydration and/or that TMAO cannot relieve binding of MB to tau. While MB retains tau in a monomeric aggregation-incompetent state in the absence of LLPS[15,77], we speculate that it could function as a molecular sticker and bridge intermolecular contacts between tau molecules under LLPS conditions, thus promoting tau LLPS and accelerating gelation of tau droplets (Fig. 8b). Further

structure–activity analysis or screening may reveal additional molecules able to modulate the LLPS propensity of tau and/or material properties of tau droplets in the future[50,78].

Does gelation of tau liquid droplets have biological implications? As a microtubule-binding protein, tau regulates the assembly of microtubules[1,2], where liquid tau droplets can nucleate microtubules and promote microtubule assembly[28]. Although the rheological properties of tau gel-like particles remain unexplored, results reported in this study show that gelation retards the mobility of tau and the recruited molecules inside the tau droplets, suggesting that the timescales for making and breaking crosslinks in the network formed by tau and interacting molecules are prolonged. Interestingly, our results demonstrated that the assembly kinetics of microtubules in the tau droplets treated with MB was only slightly slower than in the absence of MB. Similarly, in the LLPS-mediated aggregation pathway, tau fibrils grow in the aged gel-like droplets[31]. These results suggest that protein assembly within a droplet may be insensitive to the fluidity of the host droplet. It should be noted that many enzymes and regulators are involved in the microtubule assembly process[37,38], and so whether their activities are affected by gelation of tau condensates remains to be explored in the future. Furthermore, tau is involved in stress granule formation by binding to RNA, G3BP1, and TIA1[79,80]. It will also be important to investigate whether MB-tau interaction modulates the formation and the material properties of stress granules.

To summarize, we have shown that MB and its derivatives promote the LLPS of tau protein and accelerate the liquid-to-gel transition of tau droplets. These activities of MB are mediated by extensive hydrophobic and electrostatic interactions between tau and MB and are independent of MB's redox activity which has been suggested to

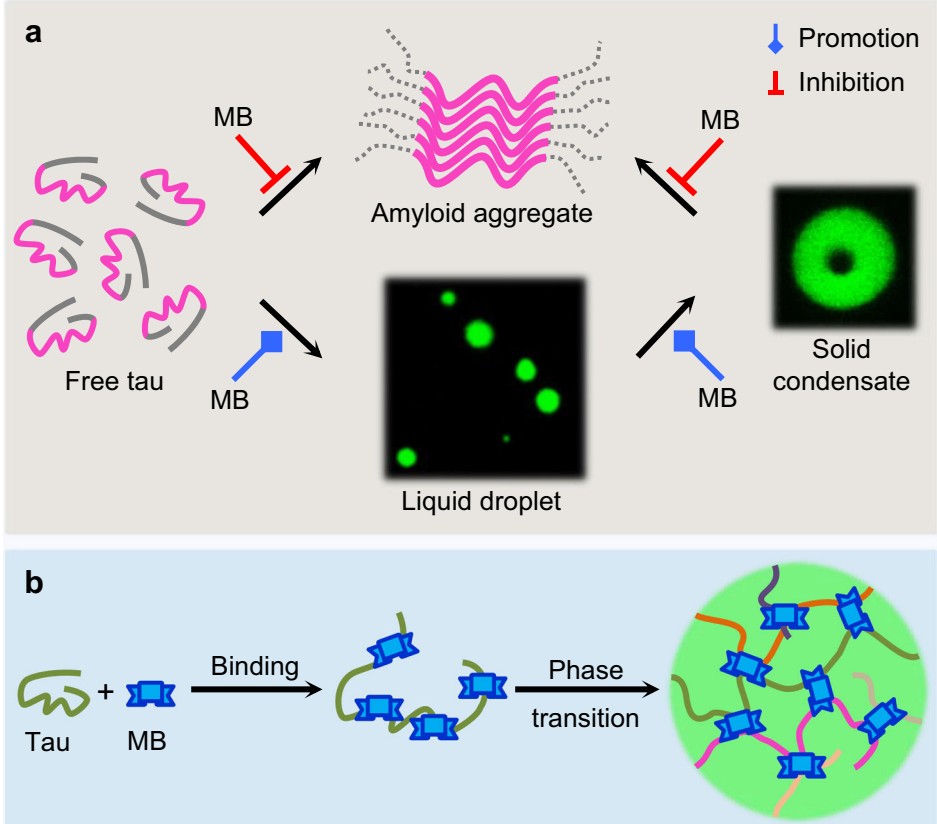

**Fig. 8 | Illustration of MB-regulated tau phase transition. a** Activity of MB in the phase transition of tau. MB inhibits aggregation of free tau and transition of gelated droplets to aggregates. In contrast, MB promotes LLPS and the liquid-to-gel transition of tau. **b** Schematic illustration of MB-tau interactions during the phase transition process. MB binds to tau, inducing slight conformational changes in tau. Upon phase separation, tau undergoes significant conformational expansion and MB may bridge interactions between tau molecules.

play a role in inhibiting tau amyloid aggregation. Rather than enhancing dehydration, MB may facilitate tau droplet gelation by introducing noncovalent crosslinks. While gelation slows down the mobility of tau and tubulin inside a droplet, droplet gelation induced by MB does not markedly impair tubulin assembly within tau droplets. We provide evidence suggesting that protein assembly within a droplet may be insensitive to the fluidity of the host droplet.

## Methods

### Expression and purification of tau variants

The deletion variants of tau were constructed by PCR using a gene encoding WT 2N4R human tau isoform (441 residues, Uniprot ID P10636) as the template. The double-Cys variants of tau were constructed as described in our previous study[35]. All genes were subcloned into a modified pET28 plasmid (Novagen) to express tau as intact protein without poly-His tag. Proteins were expressed in *Escherichia coli* BL21(DE3). Bacteria were grown at 37 °C and harvested 6 h after induction with 0.4 mM isopropyl 1-thio-β-$_D$-galactopyranoside (IPTG) at optical density 600 nm (OD600) ~ 0.8. Cell pellets were suspended in the lysis buffer (20 mM MES, 500 mM NaCl, 1 mM EDTA, 0.2 mM MgCl$_2$, 5 mM DTT, 1 mM PMSF, protease inhibitor mixture, pH 6.8) and lysed by sonication on ice. Lysates were heated at 95 °C for 20 min and cooled on ice for 20 min, and then centrifuged at 13,000 × *g* for 30 min. The supernatant was dialyzed against buffer containing 20 mM MES, 50 mM NaCl, 1 mM EDTA, 1 mM MgCl$_2$, 5 mM DTT, 0.1 mM PMSF, pH 6.8. Full-length tau, ΔNTD, and ΔCTD variants were purified by cation-exchange chromatography (SP Sepharose HP, GE Healthcare), whereas tau ΔPRD and ΔMTBD were purified by anion-exchange chromatography (Q Sepharose FF, GE Healthcare). Proteins were

further purified by size-exclusion chromatography (Superdex 75 26/600, GE Healthcare). A gene encoding enhanced green fluorescent protein (eGFP) with the A206K mutation and 6×His tag was fused to the C-terminus of tau variants to generate GFP-tau fusion proteins. GFP-tau variants were expressed in *E. coli* BL21(DE3) at 18 °C and harvested after induction with 0.4 mM IPTG at OD600 ~ 0.8 for 12 h. Cell pellets were suspended in the lysis buffer (20 mM Tris-HCl, 5 mM imidazole, 500 mM NaCl, 5 mM DTT, 10% glycerol, 1 mM PMSF, protease inhibitor mixture, pH 7.4) and lysed by sonication on ice. Lysates were centrifuged at 13,000 × *g* for 30 min. Supernatants were purified on a nickel-affinity column (Ni-NTA, Qiagen). All proteins were buffer exchanged into 50 mM Tris-HCl, 1 mM TCEP, pH 7.4 by ultrafiltration (Millipore). Protein purity was verified by SDS-PAGE and concentration was estimated by absorption at 280 nm using an extinction coefficient calculated using the ExPASy ProtParam tool (https://web.expasy.org/protparam/). The proteins were then concentrated, aliquoted, flash-frozen, and stored at −80 °C until use.

**Turbidity measurements and fluorescence microscopy imaging**
Tau proteins were incubated with or without MB (Macklin), LMTM (MedChemExpress), azure A (Sigma-Aldrich), or azure B (Sigma-Aldrich) in 50 mM Tris-HCl (pH 7.4) with 5% PEG8000 or in 20 mM HEPES (pH 7.4) at 37 °C for 10 min. Turbidity (optical density at 350 nm, OD350) was measured using a 50 μL quartz cuvette and Nanodrop 2000c (Thermo Scientific). To visualize droplets by fluorescence microscopy, unlabeled protein was mixed with GFP-fused protein at a 10:1 molar ratio. After incubation, aliquots (20 μL) were taken from the sample and dropped onto a glass slide and covered with a cover glass. Droplets were visualized using a Leica SP8 laser

scanning confocal microscope (LSCM) with a ×63 oil immersion objective at room temperature and analyzed using the Leica LAS X software.

## Concentration measurements after phase separation

The saturation concentrations of tau under different concentrations of MB, azure A, or azure B were measured using centrifugation[44]. When phase separation reaches equilibrium, the concentration of tau in the dilute phase is equal to the saturation concentration[81]. A mixture of tau and GFP-tau fusion protein at a 10:1 molar ratio was used to prepare the phase-separated samples. After incubation at 37 °C for 30 min, the samples were centrifuged at $16,000 \times g$ for 10 min. The GFP fluorescence intensities of the supernatants were measured using a microplate reader (synergy H1, BioTek) with an excitation wavelength of 480 nm and an emission wavelength of 510 nm. The concentration of tau was then calculated from the fluorescence intensities based on standard curves prepared by a mixture of tau and GFP-tau fusion protein at a 10:1 molar ratio and different concentrations of MB, azure A, or azure B without phase separation. The concentrations of MB, azure A, and azure B in the supernatants were measured using absorbance at 600 nm (Nanodrop 2000c, Thermo Scientific). To estimate the concentrations of tau, MB, azure A, and azure B in the condensates, a small volume of condensates (1 µL) was dissolved with 99 µL buffer. The fluorescence intensity and absorbance at 600 nm were measured and the concentrations of tau, MB, azure A, and azure B were calculated by multiplying the dilution factor.

## Microtubule assembly and fluorescence microscopy imaging

Microtubule assembly assays were performed in the presence of pre-formed tau (25 µM) droplets in 25 mM HEPES (pH 6.9), 2 mM MgCl₂, 0.5 mM EGTA, 1% PEG8000. Porcine brain tubulin (Cytoskeleton) was added to tau droplets at a concentration of 5 µM and incubated on ice for 15 min with or without 100 µM MB. Microtubule assembly was triggered by adding 1.0 mM GTP and incubating the reaction mixture at 37 °C. To quantify the assembly kinetics, 1 µM DAPI was added into the reaction mixture and the fluorescence of DAPI was measured using a microplate reader (synergy H1, BioTek) with an excitation wavelength of 360 nm and an emission wavelength of 450 nm. To visualize microtubule growth within tau droplets, GFP-tau and AlexaFluor647 (Thermo Fisher Scientific) labeled tubulin were mixed with unlabeled protein at a 1:10 molar ratio. Samples were removed from the 37 °C incubator at indicated time points and visualized using a Leica SP8 LSCM with a ×63 oil immersion objective.

## FRAP of droplets

FRAP experiments were performed using a Nikon A1 HD25 LSCM with a ×60 water objective and analyzed using NIS-Elements 5.21. A circular region of interest within a droplet was bleached and three droplets were selected for each condition. Each FRAP experiment involved four prebleaching frames, one or two flashes of bleaching, and at least 25 postbleaching frames. Individual FRAP traces were normalized such that the mean of prebleach values was set to 1 and the first postbleach value was set to 0.

## ThT assay and fluorescence microscopy imaging

The amyloid aggregation kinetics of tau was monitored by the fluorescence of ThT. In the assay, tau protein, 10 µM ThT (Sigma-Aldrich), and different concentrations of MB were mixed in 50 mM Tris-HCl (pH 7.4) with 7.5% PEG8000 or in 20 mM HEPES (pH 7.4) without PEG8000. The final concentrations of tau in Tris buffer and HEPES buffer were 10 and 15 µM, respectively. Tau aggregation was induced by the addition of 5 µM heparin (MedChemExpress). The mixture was then incubated at 37 °C without agitation (for Tris buffer) or with agitation at 250 rpm (for HEPES buffer). The fluorescence of ThT was measured using a

microplate reader (synergy H1, BioTek) with an excitation wavelength of 440 nm and an emission wavelength of 490 nm.

To detect the amyloid formation within tau droplets, the ThT fluorescence imaging was carried out on a Nikon A1 LSCM. A mixture of 10 µM tau, 5 µM heparin, and 20 µM ThT in 50 mM Tris-HCl (pH 7.4), 7.5% PEG8000 with or without 100 µM MB were incubated in a 96-well glass bottom plate (Cellvis) at 37 °C. Parafilm was used to cover the plate to avoid evaporation. The DIC and fluorescence images were taken at different times of incubation (0, 18, and 36 h) with a 405 nm laser as the excitation source.

## ¹⁵N-labeled tau protein preparation and NMR spectroscopy

¹⁵N-labeled WT tau protein was prepared using a previously reported method[82]. The gene for WT tau was cloned into a pNG2 vector (Novagen) which was used to transform BL21(DE3) E. coli cells. Cells were grown to an OD600 of 0.8 in Luria broth at 37 °C, after which they were pelleted and resuspended in M9 minimal medium (6 g/L Na₂HPO₄, 3 g/L KH₂PO₄, 0.5 g/L NaCl, 4 g/L glucose, pH 7.0) supplemented with 1 g/L ¹⁵N-ammonium chloride as the sole nitrogen source. Protein expression was induced by addition of IPTG to 0.5 mM. Bacterial cells were harvested by centrifugation then resuspended in ice-cold lysis buffer (20 mM MES, 1 mM EGTA, 0.2 mM MgCl₂, 5 mM DTT, 1 mM PMSF, 1 mg/mL lysozyme, 10 µg/mL DNAse, pH 6.8) and lysed by French press. To the cell lysate, 500 mM NaCl was added, after which the mixture was boiled to denature proteins. Cell debris, DNA, and precipitated proteins were removed by centrifugation ($12,700 \times g$, 40 min, 4 °C). Residual nucleic acids were precipitated by incubating the supernatant with 20 mg/mL streptomycin, and the pellet was separated by centrifugation. To collect a tau-enriched protein pellet, the supernatant was incubated with 0.36 g/mL ammonium sulfate. The protein pellet was resuspended and dialyzed into a low-salt buffer (20 mM MES, 1 mM EDTA, 2 mM DTT, 50 mM NaCl, pH 6.8). Tau was purified from the protein mixture after one round of cation exchange chromatography (MonoS 10/100, 2 mL/min using a buffer gradient of 50 mM to 1 M NaCl in 20 mM MES, 1 mM EGTA, 2 mM DTT, pH 6.8), and two rounds of size exclusion chromatography (Superdex 75 26/600, 2 mL/min in 25 mM HEPES, 100 mM KCl, 5 mM MgCl₂, 1 mM TCEP, pH 7.4). The purified protein was concentrated, flash-frozen in aliquots, and stored at −80 °C until use in NMR experiments. Tau protein was buffer exchanged into the appropriate NMR buffer using 7-kD MW cut-off Zeba spin desalting columns (Thermo Fisher Scientific) and concentrated when needed using 3-kD MW cut-off Viva Spin (Sartorius) centrifugal concentrators. Protein concentration was determined by bicinchoninic acid assay (Pierce).

## NMR spectroscopy and DIC microscopy

To observe the effect of MB, azure B, or PEG8000 on backbone H-N pairs of tau protein, NMR titrations were performed by collecting the 2D ¹H-¹⁵N SOFAST-Heteronuclear Multiple Quantum Coherence (HMQC) spectra[57] of ¹⁵N-labeled tau under various conditions. Spectra were acquired at 5 °C (278.15 K) on a Bruker Avance Neo 800.53 MHz spectrometer equipped with a triple-resonance cryoprobe CP2.1 TCI 800S6 H/C/N/D-05 Z XT (Bruker BioSpin GmbH, Germany) using TopSpin version 4.0.7 (Bruker BioSpin GmbH, Germany). The recycle delay was 200 milliseconds and the number of scans collected was 64. The spectral width was set as 10.77 ppm for the ¹H dimension and 24.50 for the ¹⁵N dimension. The center of the ¹H dimension (O1P) was set as 4.696 ppm and the center of the ¹⁵N dimension (O3P) was 118.250 ppm. The SOFAST excitation frequency corresponding to tau amide protons was 7.85 ppm and the excitation bandwidth was 2.30 ppm. NMR titration samples were prepared in LLPS-promoting buffer (50 mM Tris buffer at pH 7.4), with 90% H₂O/10% D₂O.

To determine which tau residues are involved in MB- and azure B-induced LLPS, NMR samples were prepared on ice with ¹⁵N-labeled tau (18 µM), MB or azure B (360 µM), and with or without 5% PEG8000.

Each sample was subjected to a strictly timed workflow of NMR experiments and DIC imaging, as follows. First, the tau:MB/azure B sample without PEG was incubated at 37 °C for 10 min (same as incubation time for turbidity analysis and fluorescence microscopy imaging), followed by 10 min at 5 °C (corresponding to the SOFAST-HMQC temperature), then imaged with DIC at room temperature within 5 min or subjected to SOFAST-HMQC with a dead time of 5 min between setting up the NMR experiment and starting spectral acquisition. After the first SOFAST-HMQC spectrum was recorded, 5% PEG8000 was added and the sample was split again into two parts: one for another round of DIC imaging, and the other portion for SOFAST-HMQC (with 10-min incubation at 37 °C then 10-min incubation at 5 °C, followed by imaging or spectral acquisition).

All spectra were processed using TopSpin version 3.6.2 (Bruker BioSpin GmbH, Germany) and analyzed using NMRFAM-SPARKY[83] version 1.470. Peak amplitude ratios ($I/I_0$) and chemical shift perturbations (CSP) were determined for assigned tau residues. Assignments of tau amide $^1H/^{15}N$ pairs were transferred from a previous study where tau spectra were likewise acquired at 278.15 K[62]. The $I/I_0$ ratio was calculated by dividing the peak amplitude ($I$) of tau amide peaks in the presence of MB/azure B by the corresponding peak intensity ($I_0$) in the absence of the compound. Peak amplitude ratios were likewise determined to compare SOFAST-HMQC spectra in the presence and absence of 5% PEG8000. To compare MB- and azure B-induced spectral changes against those induced by 5% PEG8000, the peak amplitude ratios from tau-MB and tau-azure B comparisons were divided by the maximum peak amplitude ratio for the reference tau spectrum containing 5% PEG8000 (this yields normalized $I/I_0$ values). CSPs were calculated using the equation $CSP = \sqrt{0.5 \times [(\Delta\delta H)^2 + (\Delta\delta N)^2/25]}$, where $\Delta\delta H$ and $\Delta\delta N$ correspond to the chemical shift differences of H and N. $I/I_0$ ratios and CSPs were calculated with a 3-residue averaging window.

### Ensemble FRET spectrum measurements and fluorescence microscopy imaging

The four double-Cys variants of tau were labeled with AF350 and AF488 according to the procedures described previously[35,60]. To detect the conformational changes of tau upon LLPS, a solution of unlabeled tau dopant with 0.2 μM AF350/AF488-labeled tau variants was prepared with different MB concentrations in 20 mM HEPES (pH 7.4) or 50 mM Tris-HCl (pH 7.4). To detect the effect of MB on the native tau protein, a solution of 0.2 μM AF350/AF488-labeled tau variants in the presence of different concentrations of MB was prepared in 20 mM HEPES (pH 7.4) or 50 mM Tris-HCl (pH 7.4). The fluorescence spectra of the samples were measured on a Shimadzu RF-5301PC fluorimeter with excitation at 347 nm and emission from 350 to 600 nm.

In order to determine the distribution of AF350/AF488-labeled tau variants in the LLPS state, the fluorescence imaging was carried out on a Nikon A1 LSCM equipped with a ×100 oil-immersed objective and a 96-well glass bottom plate (Cellvis). The tau droplets were prepared by incubating unlabeled tau doped with 10% AF350/AF488-labeled tau either in 20 mM HEPES (pH 7.4) without PEG8000 or 50 mM Tris-HCl (pH 7.4) with 5% PEG8000, with or without 100 μM MB.

To investigate the effect of MB on the conformations of phase-separated tau in the condensed phase, we incubated unlabeled tau doped with 4% AF350/AF488-labeled tau in 50 mM Tris-HCl (pH 7.4), 5% PEG8000, with or without 100 μM MB. A sample volume of 30 μL was dropped onto a glass slide and covered with a coverslip. The samples were visualized using a two-photon laser confocal microscope (Nikon A1R MP) using a ×25 water immersion objective. The excitation wavelength was 700 nm to excite AF350 efficiently in a two-photon manner. The fluorescence images of the tau droplets were taken in both AF350 channel and AF488 channel simultaneously. The apparent

FRET efficiencies of the droplets were calculated by dividing the intensity in the AF488 channel by the total intensity of both channels.

### Fluorescence lifetime measurements

Fluorescence lifetime measurements of AF488-labeled tau in the native state and/or in the LLPS state were performed on an FLS920 spectrometer (Edinburgh Instruments). The samples with or without 100 μM MB were excited with a 485 nm pulse laser and detected at 525 nm. The histograms of the photon arrival time were recorded until the peak counts reached 5000. The time-resolved fluorescence decay traces were analyzed with Origin2023 program and fitted to a single-exponential function to obtain the characteristic decay time.

### TEM

Samples for TEM were prepared as those for ThT assay without addition of ThT. After 18 h and 36 h, 5 μL of each sample was added onto 230-mesh carbon-coated EM copper grids and allowed to adsorb for 1 min. To observe the formation of tau droplets at the beginning, the samples were directly adsorbed onto carbon-coated EM grids for 1 min and fixed by adding 4% (w/v) paraformaldehyde for 3 min. Excess protein was removed by washing the EM copper grids three times with fresh buffer (50 mM Tris-HCl, pH 7.4). The grids were then washed with 5 μL pure water and stained with 5 μL 3% uranyl acetate for 30 s. The images were taken on a 120 kV Tecnai Spirit electron microscope and analyzed using SerialEM Version 4.0.9.

### Cell toxicity experiments

Human neuroblastoma (SH-SY5Y) cells (CRL-2266, ATCC) were cultured in Dulbecco's modified Eagle's medium (Gibco), supplemented with 10% fetal bovine serum (Gibco), 100 μg/mL streptomycin and 100 U/mL penicillin, and incubated at 37 °C in a humidified environment containing 5% $CO_2$. To determine the effects on cell viability, the cell number was quantified using a standard colorimetric 3-(4,5-dimethylthiazol-2-yl)-2,5-diphenyltetrazolium bromide (MTT) assay. Cells were transferred to a 96-well plate ($1 \times 10^4$ cells/well) and allowed to attach overnight. The preparation of the tau samples for MTT assay was the same as that for ThT assay without addition of ThT. The samples were taken at different incubation time (0, 5, and 24 h) and each diluted into culture medium to reach 2.5 μM final concentration of tau. Cells were treated with the above tau samples in culture medium for 24 h. After that, 10 μL of MTT (5 mg/ml stock in PBS) was added to the culture medium in each well and the cells were further incubated for 4 h at 37 °C. Finally, the culture medium was discarded carefully and 150 μL of DMSO was added per well to dissolve the purple formazan crystals present. Absorbance was measured using a microplate reader (Molecular Devices) at 490 nm.

### Measurement of aspect ratios of fused droplets

Tau droplets were formed by mixing 5 μM tau protein (R&D Systems) in 25 mM HEPES (pH 7.4), 150 mM KCl, 1 mM DTT, 15% PEG6000 and MB (ChemScene) at the indicated concentrations. A 1 μL sample was transferred into a cover glass chamber and observed in the optical tweezers instrument (see below). Two tau droplets of similar sizes (2–3 μm in diameter) were trapped by two separate optical traps. The two tau droplets were slowly brought to touch each other with the help of a steerable mirror that controls one of the laser beams in a dual-trap optical tweezers instrument (1064 nm YAG laser, CW 4 W, BL-106C, Spectra physics, Mountain View, CA)[84]. The laser traps were turned off as soon as the two droplets touched each other, which was monitored for several seconds in a home-built video microscope equipped with a Nikon objective (CFI Plan Apo ×60 water immersion, NA 1.2 MRD07602) and a CCD camera (Watec-902H2, 1/2 INCH B/W high resolution CCD Camera) in a time window ranging from 10 min to 3 h. The aging time was started when PEG was added to the system.

After the fusion was complete, the length and width of the two touching droplets were measured using the image analysis software ImageJ[85] to calculate the aspect ratio using the formula: aspect ratio = length/width.

## Docking simulation

We studied the binding poses of the ligands (MB, azure A, azure B, and TMAO) with tau through docking simulation. We selected 100 representative tau structures from all-atom simulations of the full-length tau as the receptors[86]. Considering the large size of the tau protein, we divided the protein into ten fragments and for each docking study, a fragment was placed at the center of the docking box. The box size was adjusted so that any atom in the fragment had a minimum distance of 20 Å to the borders of the box (Supplementary Fig. 21). In total, we conducted 1000 docking simulations for each compound using AutoDock Vina[87]. For each docking simulation, only the best docking pose was selected for post-analysis.

## Statistics and reproducibility

All experiments were repeated at least twice except the NMR experiments which were performed once. Data are presented as mean ± SD. Unpaired two-sided Student's $t$-test was performed to compare data from two conditions using Origin 8. Sample sizes and statistical details are indicated in figure legends.

## Reporting summary

Further information on research design is available in the Nature Portfolio Reporting Summary linked to this article.

## Data availability

The data are available within the article, Supplementary Information, or Source data file, and available from the corresponding authors upon request. The raw NMR spectra files will be available upon request. Source data are provided with this paper.

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

## Acknowledgements

This work was supported by National Natural Science Foundation of China (#32000883 to M.G., #31920103011 to S.P., and #32171443 to S.W.), China Postdoctoral Science Foundation (#2022TQ0357 to J.W.), the CAS Center of Excellence in Biomacromolecules (S.P.), Hubei University of Technology (Y. Huang, M.G., and Z.S.). M.Z. was supported by the European Research Council (ERC) under the EU Horizon 2020 research and innovation program (grant agreement No. 787679). Y. Huang thanks Zhe Hu (Huazhong Agricultural University) for his help with the FRAP experiments and Chengdong Huang (University of Science and Technology of China) for sharing a plasmid encoding 2N4R tau. L.-M.R., E.G., and M.Z. thank Maria-Sol Cima-Omori for preparation of $^{15}$N-labeled tau.

## Author contributions

Y. Huang and M.G. conceived the project. H.Y., P.L., Y. Han, and Y. Huang carried out the FRAP and fluorescence microscopy imaging. H.Y., Y.L., and Y. Huang carried out turbidity measurements. H.Y., Y. Han, P.L., and J.W. produced the protein samples. P.P. carried out aspect ratio measurements. J.W. carried out FRET, TEM, and cell experiments. L.-M.R. and E.G. carried out NMR experiments and DIC microscopy imaging. H.Y., Y. Han, and Y. Huang carried out ThT assay and tubulin assembly experiments. V.H.M. performed docking simulations. All authors analyzed the data and discussed the results. Y. Huang and M.G. wrote the first draft with contributions from all the authors. Y. Huang, H.M., L.-M.R., M.Z., J.W., Z.S., S.P., S.W., and M.G. edited the manuscript. Y. Huang, H.M., M.Z., J.W., Z.S., S.W., and M.G. supervised and directed the project.

## Competing interests

The authors declare no competing interests.
