## [Peer Review file · Nature Communications]

REVIEWER COMMENTS

Reviewer #1 (Remarks to the Author):

The manuscript by Huang et al. explores the question how a small molecule inhibitor of tau fibrillization influences phase separation. The authors use phase separation and fibrillization assays, and also use alternative biophysical approaches to characterize the interaction of the small molecule with tau. They conclude that loss of dynamics of the phase separated state is not necessarily coupled to enhanced fibrillization. This is an interesting result that is important to report in my opinion given the unfortunate convolution of "gelation" and "fibrillization" in the phase separation literature. However, the manuscript suffers from several shortcomings. It falls prey to the temptation of using the same suggestive language ("gelation") that resulted in the convolution in the first place, without their being a physical basis for this interpretation. Furthermore, some data are over-interpreted or missing controls. Hence, while I maintain that this is a very worthwhile direction, I consider the manuscript too preliminary for publication in Nature Communications.

Main weaknesses:

1. The manuscript equates loss of dynamics of condensates with gelation. What is the evidence that the aged condensates are gels but that nascent condensates are not? Recent physics-based explanations of phase separation describe even nascent condensates as "micro-gels" (PMID: 35675815).
2. It is unclear whether the addition of methylene-blue (MB) indeed enhances dynamical arrest of condensates or whether it just increases the driving force for phase separation by introducing additional crosslinks. If the first possibility is the case, dynamical arrest and fibrillization would be separated functionally; in the other case, protein molecules would simply be sequestered in the dense phase and unavailable for fibrillization. Given the claims in the paper, it is essential that these two possibilities are dissociated. This will not be possible without measuring saturation concentrations of tau under the different conditions. Currently, the authors use turbidity as a stand-in for the driving force for phase separation, but this is a poor measure given that the same dense phase volume fraction can result from different size distributions of scattering particles. The authors acknowledge that the size distributions can change – they say that Azure B prevents fusion of condensates. This effect can also explain why microscopy images look similar in terms of droplets size and number, but the turbidity is different. The authors erroneously attribute this to changes in the interaction between MB and tau. If that was the case, the driving force for phase separation should change.
3. PEG is a molecular crowder with effects on phase-separation and aggregation. The authors should explain the rationale behind using PEG, when phase-separation is achievable without it. Switching between PEG and no PEG conditions for inducing phase-separation in different experiments, without providing any explanation, is difficult to follow. In the droplet ageing experiments (Fig. 1) FRAP has been done in the absence of PEG, but for the optical tweezers experiment for monitoring the aspect ratio of droplets they have again used PEG. What is the reason for this and will this not result in changes in dynamics between the two methodologies?
4. In the NMR experiments, the chemical shift perturbations appear to be most prominent for the Δ NTD compared to the rest. The turbidity increase is smallest for Δ NTD in the presence of MB as well. This could possibly suggest a more important role for the NTD in phase-separation, but the authors give the impression that the driving force for phase separation is distributed evenly across the sequence. It is also not clear whether tau is in a phase-separated state in the absence and presence of MB. If yes, they should also show controls of MB with tau in a non-phase-separated state as they

claim that the mode of interaction between tau and MB and hence its inhibitory mechanism is different in a non-phase separated versus a phase-separated state.

5. The FRET experiments are impossible to interpret without knowing which fraction of tau is in the dense phase; presumably only a small fraction. Is it reasonable then that the effect on FRET is large, or does this suggest that this is an artifact, e.g. due to a change in lifetime of one of the fluorophores? By how much do the dimensions of tau have to change to result in the reported change in FRET? This experiment requires careful controls, including the effect of MB on non-phase-separated tau. They should also show proper controls with the two individual dyes in the dense-phase to make sure that the dye fluorescence is observable in the dense-phase.

Minor points:

6. The authors characterize the effect of MB on phase-separation and material properties of tau droplets but they fail to test whether MB enters the droplets also this is an important part of their model. This should be tested.

7. The authors comment (Page 8, Line 217) that gelation of droplets can retard the fusion of small droplets into larger ones. Therefore, tau droplets in the presence of azure B were smaller than those in the presence of MB and azure A. This could be true but the conclusion cannot be made based on the presented data if they have not compared the phase-separation propensities between tau with MB, Azure A and Azure B, as smaller droplets could also arise from the lower phase-separation propensity in the presence of azure B.

8. In panel Fig. 4b for the effect on phase-separation on deletion mutants, the authors have only shown turbidity assay without corroborating phase separation with microscopy images.

9. The explanation that HD affects specifically hydrophobic interactions is not accurate. It is a highly polar molecule.

10. Incorrect statement (Page 12, Line 323) - The result shows that the distance between the labeling pair "decreases" upon MB induced phase separation - it should be "increases" if the conformation is becoming more expanded.

11. The presence of PEG in ThT assay could have its own effect. As the authors are able to induce phase separation without PEG, they should do a ThT assay without PEG in the absence and presence of MB to analyze its effect on aggregation.

12. They have done all the droplet assays without heparin but used heparin in the ThT assay. If heparin is absolutely necessary for mediating tau aggregation, they should show a control for phase separation of tau in the presence of heparin to show that it forms droplets in a manner similar to without heparin.

13. For the TEM images, the 0-hour time point is missing. Were they able to observe any droplets at 0 h in the phase separated conditions?

14. Instead of just relying on the ThT assay to make the point about fibrils and doing TEM of the ThT assay samples, which will have contributions from the dilute as well as the dense-phase, they should also try to directly monitor the fibrillization going on in droplets to really make the point that fibrils are not forming in the aging droplets and compare it to a control without MB showing amyloid fibril formation in the droplets.

Reviewer #2 (Remarks to the Author):

In this study, Huang and co-authors investigated the role of a Tau aggregation inhibitor: methylene blue (MB), in regulating the liquid droplet formation and liquid-to-gel transition of Tau protein. Their results show that MB promotes tau liquid-liquid phase separation and accelerates the liquid-to-gel transition of tau droplets. In addition, the effect of MB is independent of the redox activity of methylene blue, unlike their activities in inhibiting Tau aggregation. Furthermore, methylene blue inhibits the conversion of tau droplets into fibrils. These results are interesting, because it is believed that liquid-to-gel transition is an intermediate step of fibrillization. The fact that MB accelerates liquid-to-gel transition, but inhibits aggregation indicate that the mechanism or interaction of MB with Tau in these two steps are different. The paper is very well written and organized. The data presented in this manuscript sufficiently support their conclusion. However, because of the opposite effect of MB on aggregation and liquid droplet formation, I would appreciate more study to understand the mechanism of MB-Tau interaction that causes these differences.

1. The authors conducted NMR experiments to show that MB interacts with several regions in tau. However, the experiment was done in phosphate buffer and not in droplet forming condition. In order to understand the differences in the interaction that inhibits Tau aggregation and promotes Tau LLPS, the NMR experiment should be done in LLPS buffer and compare with the results obtained in phosphate buffer.
2. The authors investigated several MB derivatives and found azure B was much weaker than MB in inducing turbidity of Tau, but stronger in inducing gelation of tau droplets. This is interesting and worth further investigation. Since turbidity is affected by many factors (size, number, et. al.,) of the droplets, it is not the best method to quantify phase separation propensity and should be complemented by another method. Sedimentation assay could be used to determine the critical concentration at which LLPS occurs.
3. According to Ref 51, the IC50 of MB and azure B in inhibiting Tau aggregation is the same. But their effect on Tau phase separation is different, therefore, NMR study of Tau structure in the presence of MB and azure B might be able to tell which interactions are responsible for the inhibition activity and which interactions are responsible for the promoting activity.
4. The authors showed that a Tau liquid-to-gel transition inhibitor TMAO, at a concentration up to 0.2 M, could not mitigate the liquid-to-gel transition of tau droplets in the presence of MB, azure A or azure B. Some mechanistic insight of this result would be appreciated. Does TMAO engage the same regions as MB? Are there competitive binding?
5. From the FRET experiments, the authors concluded that conformational expansion of tau in the droplets results from phase separation rather than from MB binding. Based on the data presented, it is clear that binding of MB in non-LLPS buffer does not change the conformation of Tau. However, since interaction between MB and Tau in non-droplet forming condition, is different than in droplet forming condition, we can't rule out the possibility that in droplets, binding of MB enhances the conformational expansion.
6. For figure 7, the authors concluded that gelation of tau droplets induced by MB slightly retards the assembly kinetics of tubulin within tau droplets without affecting the final appearance of microtubule bundles. The conclusion is drawn by visual observation of the fluorescence images. I would appreciate that the authors to quantify the assembly kinetics and test the significance of the differences with/without MB.

Response to Reviewer #1

1. Reviewer: *“The manuscript equates loss of dynamics of condensates with gelation. What is the evidence that the aged condensates are gels but that nascent condensates are not? Recent physics-based explanations of phase separation describe even nascent condensates as “micro-gels” (PMID: 35675815).”*

Thanks for the reviewer’s comments. We referred to the suggested paper and learned that gelation involves percolation transition. However, it is difficult to determine the network structure within condensates experimentally. We applied FRAP to measure the mobility of molecules within the droplets and optical tweezers to measure the fusion kinetics of droplets. Our results showed that tau droplets are highly dynamic and undergo fusion without MB, suggesting that they are in a liquid-like state. In contrast, the dynamics of tau droplets is reduced and fusion is prohibited as the concentration of MB is increased, suggesting the occurrence of gelation. The term “gelation” has been widely adopted in recent publications to indicate loss of dynamics of condensates, such as references 65-68 in the revised manuscript. To clarify the usage of gelation, we have revised the manuscript on Page 18 as follows:

“In this study, we investigated the influence of the aggregation inhibitor MB on the phase transition behavior of tau. Our results demonstrate that MB promotes tau LLPS and decreases the dynamics of tau droplets. Such loss of dynamics of condensates has previously been termed gelation⁶⁵⁻⁶⁸.”

2. Reviewer: *“It is unclear whether the addition of methylene-blue (MB) indeed enhances dynamical arrest of condensates or whether it just increases the driving force for phase separation by introducing additional crosslinks. If the first possibility is the case, dynamical arrest and fibrillization would be separated functionally; in the other case, protein molecules would simply be sequestered in the dense phase and unavailable for fibrillization. Given the claims in the paper, it is essential that these two possibilities are dissociated. This will not be possible without measuring saturation concentrations of tau under the different conditions. Currently, the authors use turbidity as a stand-in for the driving force for phase separation, but this is a poor measure given that the same dense phase volume fraction can result from different size distributions of scattering particles. The authors acknowledge that the size distributions can change – they say that Azure B prevents fusion of condensates. This effect can also explain why microscopy images look similar in terms of droplets size and number, but the turbidity is different. The authors erroneously attribute this to changes in the interaction between MB and tau. If that was the case, the driving force for phase separation should change.”*

Thanks for the reviewer’s comments. We agree with the reviewer that it’s inadequate to analyze phase separation using turbidity only. Following the reviewer’s suggestion, we measured the saturation concentrations of tau in the presence of different concentrations of MB, azure A, or azure B using

centrifugation. We found that the saturation concentration of tau decreases markedly as the concentrations of MB or azure A are increased (Figures 1d and 3d), indicating that MB and azure A enhance the capacity of tau to undergo phase separation. In contrast, the saturation concentration of tau showed marginal changes as the concentration of azure B is increased (Figure 3d), indicating the minimal effect of azure B on the capacity of tau to undergo phase separation. The saturation concentration measurement together with the FRAP analysis suggest that azure B primarily enhances dynamical arrest of tau condensates whereas azure A and MB enhance dynamic arrest of tau condensates as well as increase the driving force for tau to undergo phase separation. We have revised the manuscript to discuss the new data as follows:

On Page 4:

“...To assess the effect of MB on the phase separation propensity of tau, we determined the saturation concentration of tau (C_{sat}) under different MB concentrations using centrifugation⁴⁴. As shown in figure 1d, C_{sat} decreased with MB concentration, suggesting that MB enhances the capacity of tau to undergo phase separation.”

On Page 8:

“...Saturation concentration measurements showed that azure A has similar effects as MB on reducing C_{sat} (Fig. 3d). In contrast, C_{sat} only changed marginally as the concentration of azure B was increased, indicating that azure B enhances dynamic arrest of tau droplets with minimal effect on the capacity of tau to undergo phase separation.”

3. Reviewer: “PEG is a molecular crowder with effects on phase-separation and aggregation. The authors should explain the rationale behind using PEG, when phase-separation is achievable without it. Switching between PEG and no PEG conditions for inducing phase-separation in different experiments, without providing any explanation, is difficult to follow. In the droplet ageing experiments (Fig. 1) FRAP has been done in the absence of PEG, but for the optical tweezers experiment for monitoring the aspect ratio of droplets they have again used PEG. What is the reason for this and will this not result in changes in dynamics between the two methodologies?”

Thanks for the reviewer’s comments. We agree with the reviewer that PEG is a molecular crowding agent with effects on phase separation and aggregation, which has been extensively used in *in vitro* studies to mimic the crowding environment in the cell. In this study, we performed most of the LLPS experiments both in the presence of PEG (Tris buffer) and in the absence of PEG (HEPES buffer), in order to confirm our finding that the regulatory activity of MB on tau LLPS is an intrinsic property of MB. The consistent results of turbidity assay, FRAP, FRET, ThT assay, and functional and cytotoxicity assays were obtained under both conditions and have been shown either in the main text or SI. The only exception is the optical tweezers experiments. In the optical tweezers experiment, PEG was used with HEPES buffer in order to enhance the LLPS of

tau. We tested various concentrations of tau and PEG for LLPS, and found that below 10% PEG the droplets formed were too small to evaluate their fusion promptly using optical tweezers. Thus, we selected 15% PEG and 5 μ M tau to evaluate their fusion, which is the most suitable conditions tested for the optical tweezers experiments. We believe that although addition of PEG may vary the absolute value of certain properties of the droplet such as FRAP recovery time and fusion time in Figure 2, the trend we observed with different MB concentrations should not be affected since at each MB concentration the PEG concentration is fixed. As shown in Figure 2, the trends of FRAP recovery and fusion kinetics are consistent with each other, which supports that inducing tau droplet gelation is an intrinsic property of MB. To clarify the PEG usage in optical tweezers experiments, we have added a statement in the legend of Figure 2 as follows:

“...To obtain droplets of suitable size for fusion kinetics measurements, PEG was added to facilitate tau droplet formation.”

4. Reviewer: *“In the NMR experiments, the chemical shift perturbations appear to be most prominent for the Δ NTD compared to the rest. The turbidity increase is smallest for Δ NTD in the presence of MB as well. This could possibly suggest a more important role for the NTD in phase-separation, but the authors give the impression that the driving force for phase separation is distributed evenly across the sequence. It is also not clear whether tau is in a phase-separated state in the absence and presence of MB. If yes, they should also show controls of MB with tau in a non-phase-separated state as they claim that the mode of interaction between tau and MB and hence its inhibitory mechanism is different in a non-phase separated versus a phase-separated state.”*

Thanks for the reviewer’s comments. We have redone the NMR experiments under phase separation conditions for a closer comparison between the phase separation analysis and the NMR binding profile. Under these conditions (Figures 4e and S14h) tau is in a phase-separated state.

This time, we interpreted the I/I₀ profile for the purpose of determining which domains in tau preferentially bind to MB, because under LLPS conditions the spectral quality is too low (most peaks are broadened beyond detection) to achieve a reliable chemical shift perturbation analysis.

The procedure for our NMR experiments done in tandem with DIC microscopy is as follows (as written in the revised methodology): “...To determine which tau residues are involved in MB- and azure B-induced LLPS, NMR samples were prepared on ice with ¹⁵N-labeled tau (18 μ M), MB or azure B (360 μ M), and with or without of 5% PEG8000. Each sample was subjected to a strictly timed workflow of NMR experiments and DIC imaging, as follows. First, the tau:MB/azure B sample without PEG was incubated at 37 °C for 10 min (same as incubation time for turbidity analysis and fluorescence microscopy imaging), followed by 10 min at 5 °C (corresponding to the SOFAST-HMQC temperature), then imaged with DIC at room temperature within 5 min or subjected to SOFAST-HMQC with a dead time of 5 min between setting up the NMR

experiment and starting spectral acquisition. After the first SOFAST-HMQC spectrum was recorded, 5% PEG8000 was added and the sample was split again into two parts: one for another round of DIC imaging, and the other portion for SOFAST-HMQC (with 10-min incubation at 37 °C then 10-min incubation at 5 °C, followed by imaging or spectral acquisition).”

With these new experiments, it is now clearer that both NTD and CTD regions (and a part of the MTBD) are involved in binding of MB under LLPS conditions (Figure 4f). From our comparison in Figure 4f (black versus magenta bar plots), it is evident that the I/I_0 profile for tau-MB (magenta) is reminiscent of that of tau-PEG (black) in regions around residue ~200 in the PRD, i.e. this region is the closest to a 1:1 correspondence between black and magenta profiles, however I/I_0 for other domains of tau (NTD, CTD, part of MTBD) in the presence of MB remain much lower than the counterpart with 5% PEG 8000.

We further explain the results on Pages 10-11 as follows:

“ ^1H - ^{15}N SOFAST HMQC spectra⁵⁷ of ^{15}N -labeled WT tau protein were recorded in the absence and presence of MB (Fig. 4d and S14b). The NMR spectra were recorded at 5 °C to minimize loss of ^1H signal due to solvent exchange⁵⁸. Under these conditions, tau did not form droplets in the absence of MB (Fig. 4e). Addition of MB resulted in the formation of a biphasic sample consisting of suspended tau droplets in a dispersed phase of tau (Fig. 4e). Quantification of the changes in peak intensities upon MB addition revealed an overall reduction in NMR signal intensities involving all domains of tau (Fig. 4f and S14f). A reduction in NMR signal intensity in relation to protein phase separation might arise from the slower tumbling of the protein in the droplet phase and associated shortened transverse relaxation times⁵⁹. In addition, exchange between an NMR-visible state of the protein (i.e. monomeric tau in the dispersed phase) and an NMR-invisible state (i.e. droplets or aggregates) potentially contributes to the reduction in signal. In the case of protein-ligand binding, NMR line broadening may also occur due to exchange between free and ligand-bound states of the protein in the intermediate exchange regime. To assess the impact of droplet formation without MB binding on the NMR spectrum of tau, we collected the ^1H - ^{15}N SOFAST HMQC spectrum of tau in the presence of 5% PEG8000 (Fig. 4d and S14a). PEG induced tau droplet formation (Fig. 4e) and led to ~40% peak amplitude reduction (Fig. 4f and S14d) as well as chemical shift perturbations (Fig. S14e) across all tau domains, consistent with the formation of a slow-tumbling NMR-invisible state (i.e. droplet phase). Notably, the intensity ratio profile for the tau-PEG mixture does not completely correspond to that of the tau-MB sample (Fig. 4f). The prominent lack of overlay for the entire NTD and CTD regions suggest that these domains are the most affected by MB binding. A patch of the MTBD (residues ~330–350) and some residues in the PRD also appear to bind to MB. We also collected the ^1H - ^{15}N SOFAST HMQC spectrum of tau in the presence of azure B (Fig. 4d and S14c). Comparing the intensity ratio profiles for tau-MB and tau-azure B (Fig. 4f), we observed that the profiles are mostly consistent, and that I/I_0 attenuation due to MB is more pronounced than that due to

azure B except for short stretches of residues such as those around ~200 in the PRD and ~380–400 in the CTD, suggesting that interaction with these residues may be responsible for the different effects of MB and azure B on inducing tau LLPS and droplet gelation. Addition of PEG8000 together with MB or azure B resulted in severe attenuation of the NMR peak intensities due to enhancement of phase separation (Fig. S14g and h), which prohibited further residue-level analysis.”

The turbidity increase is smallest for Δ NTD in the presence of MB, which may be due to a dramatic loss of negative charges that are essential for tau LLPS. We discussed this on Page 10 as follows:

“Notably, the turbidity increase for tau Δ NTD was smaller than for WT tau and the other deletion variants, which may result from a dramatic loss of negative charges that are essential for tau LLPS.”

5. Reviewer: *“The FRET experiments are impossible to interpret without knowing which fraction of tau is in the dense phase; presumably only a small fraction. Is it reasonable then that the effect on FRET is large, or does this suggest that this is an artifact, e.g. due to a change in lifetime of one of the fluorophores? By how much do the dimensions of tau have to change to result in the reported change in FRET? This experiment requires careful controls, including the effect of MB on non-phase-separated tau. They should also show proper controls with the two individual dyes in the dense-phase to make sure that the dye fluorescence is observable in the dense-phase.”*

Thanks for the reviewer’s comments. According to the reviewer’s comments, we first carried out fluorescence imaging of the AF350/AF488 dual labeled tau under LLPS conditions. Since there is no suitable laser for excitation of the AF350 dye provided on the commercial confocal microscope, we used 488 nm laser as the excitation source to detect the localization of AF350/AF488 dual-labeled tau. The imaging results indicated that the labeled protein was highly enriched in the condensed phase. The imaging results are shown in Figure S17.

Based on our data in previous Figure 5, it has already been shown that binding of 20 μ M MB in non-LLPS state does not induce significant conformational changes of tau. In order to further address the reviewer’s comments, we detected the effect of MB up to 100 μ M on AF350/AF488-labeled tau under non-phase-separating conditions. Different from that in Figure 5b and 5e, here 200 nM AF350/AF488-labeled tau (without unlabeled tau) was used, where the LLPS would not occur due to low protein concentration. The FRET signals showed a slight decrease with increasing concentrations of MB above 20 μ M (Figure S18). However, the FRET changes induced by MB were much less than in the LLPS state shown in Figure 5b and 5e. We further carried out FRET imaging of AF350/AF488 dual-labeled tau in the LLPS state, using a two-photon laser scanning confocal microscope with 700 nm laser as the light source for excitation of AF350 (Figure 5c). The results showed that upon addition of MB, the ratio of AF488/AF350 intensity within the droplets decreased, indicating the decrease in

FRET efficiency within the droplets (Figure 5d). Therefore, the bulk FRET changes are mainly attributed to the conformational changes of tau in the condensed phase. However, it is impossible to tell the detailed dimension changes of tau based on the bulk FRET efficiency because it is just an apparent value that cannot be accurately converted to distance.

In order to check whether the FRET decrease under LLPS conditions is due to conformational expansion of proteins in the condensed phase or an artifact such as quenching of the acceptor (AF488), we measured the lifetime of AF488-labeled tau (Table S2). Although the presence of PEG can cause a decrease in the AF488 lifetime, there is no significant change in the AF488 lifetime upon addition of MB that enhances LLPS, demonstrating that the FRET decreases are caused by the conformational expansion of tau in the LLPS state rather than quenching of the acceptor dye.

We further discussed the results on Page 13 as follows:

“...The AF350/AF488-labeled tau was enriched in the condensed phase upon LLPS (Fig. S17). In Tris buffer, we observed significant FRET signals for all four tau variants in their native state (Fig. 5b, black lines), indicating that AF350 and AF488 were within the distance that can undergo FRET. This is consistent with the paper-clip model of the native tau conformation^{61,62}. Incubation of tau with MB in the non-LLPS state resulted in slight changes in the apparent FRET signals (Fig. 5b, green lines, and Fig. S18a), suggesting that MB binding perturbed the conformations of tau. This is consistent with the above results from NMR and docking, which indicate extensive interactions between MB and tau (Fig. 4). Inducing tau phase separation by adding PEG decreased the FRET signal of the labeled tau variants (Fig. 5b, blue lines). A further decrease in the apparent FRET signal was observed when tau was co-incubated with MB and PEG, which led to an obvious LLPS state (Fig. 5b, red lines). Two-photon microscopy images further demonstrated that the FRET decrease occurs within the droplets upon addition of MB (Fig. 5c and d). In HEPES buffer, a similar phenomenon was observed although the crowding reagent PEG8000 was not necessary for LLPS. The FRET signals of dual-color labeled tau variants showed a slight decrease in the presence of MB in the non-LLPS state (Fig. 5c, blue lines, and S18b). A marked decrease in the FRET signal was observed when LLPS occurred (Fig. 5c, red lines). To check whether the decrease of the apparent FRET efficiency was caused by artificial quenching of the acceptor (i.e. AF488), we performed fluorescence lifetime measurements, which showed no significant change of AF488 lifetime upon addition of MB in either the native or LLPS states (Table S2). Therefore, our FRET analysis indicates that the distance between the labeling pair increases upon MB-induced LLPS especially for the NTD and CTD regions, indicating that the tau conformation changes to an expanded state, which could result from a synergistic effect of phase separation and MB binding.”

6. Reviewer: “*The authors characterize the effect of MB on phase-separation and material properties of tau droplets but they fail to test whether MB enters the*

droplets also this is an important part of their model. This should be tested.”

Thanks for the reviewer’s comments. We agree with the reviewer that testing whether MB enters the droplets is important for our work. We measured the concentrations of MB in the light phase and dense phase using centrifugation. We found that, after phase separation, the concentration of MB decreases in the dilute phase while it increases in the condensed phase compared to the initial concentration (Figure S10), indicating enrichment of MB in tau droplets. We revised the manuscript to discuss the results on Page 8 as follows:

“...By comparing the concentration of compounds in the dilute phase and the condensed phase (Fig. S10), the compounds were found to be enriched in tau droplets.”

7. Reviewer: *“The authors comment (Page 8, Line 217) that gelation of droplets can retard the fusion of small droplets into larger ones. Therefore, tau droplets in the presence of azure B were smaller than those in the presence of MB and azure A. This could be true but the conclusion cannot be made based on the presented data if they have not compared the phase-separation propensities between tau with MB, Azure A and Azure B, as smaller droplets could also arise from the lower phase-separation propensity in the presence of azure B.”*

Thanks for the reviewer’s comments. We measured the saturation concentrations of tau under different concentrations of MB, azure A, or azure B using centrifugation (Figure 3d). We found that the saturation concentration of tau decreased markedly as the concentrations of MB or azure A were increased, indicating that MB and azure A enhance the capacity of tau to undergo phase separation. In contrast, the saturation concentration of tau showed marginal changes as the concentration of azure B was increased, indicating the minimal effect of azure B on the capacity of tau to undergo phase separation. Under the same concentrations of compounds, the saturation concentration of tau for azure B is greater than those for MB and azure A, indicating that the phase separation propensity of tau in the presence of azure B is lower. We realized that the original statement may be inaccurate and has been removed from the revised manuscript. We discussed the results of saturation concentrations of tau under different concentrations of MB, azure A, or azure B on Page 8 as follows:

“...Saturation concentration measurements showed that azure A has similar effects as MB on reducing C_{sat} (Fig. 3d). In contrast, C_{sat} only changed marginally as the concentration of azure B was increased, indicating that azure B enhances dynamic arrest of tau droplets with minimal effect on the capacity of tau to undergo phase separation.”

8. Reviewer: *“In panel Fig. 4b for the effect on phase-separation on deletion mutants, the authors have only shown turbidity assay without corroborating phase separation with microscopy images.”*

Thanks for the reviewer’s comments. We have performed fluorescence microscopy imaging for the tau deletion variants under different MB

concentrations. The results are shown in Figure S12.

9. Reviewer: *“The explanation that HD affects specifically hydrophobic interactions is not accurate. It is a highly polar molecule.”*

Thanks for the reviewer’s comments. 1,6-hexanediol has been shown to perturb the permeability barrier of the nuclear pore complex and is often used to probe the material properties of droplets. Although the precise mechanism is unknown, 1,6-hexanediol is thought to mainly disrupt hydrophobic interactions in a number of LLPS studies. The related papers have been cited in the text as ref 52-55. We revised the manuscript on Page 10 as follows:

“...Since NaCl screens electrostatic interactions and 1,6-HD is thought to disrupt hydrophobic interactions⁵²⁻⁵⁵, the sensitivity of MB-induced tau droplets to NaCl and 1,6-HD suggests that MB promotes tau phase separation via both electrostatic interactions and hydrophobic interactions.”

10. Reviewer: *“Incorrect statement (Page 12, Line 323) - The result shows that the distance between the labeling pair “decreases” upon MB induced phase separation - it should be “increases” if the conformation is becoming more expanded.”*

Thank you to the reviewer. The word “decreases” has been corrected to “increases”.

11. Reviewer: *“The presence of PEG in ThT assay could have its own effect. As the authors are able to induce phase separation without PEG, they should do a ThT assay without PEG in the absence and presence of MB to analyze its effect on aggregation.”*

Thanks for the reviewer’s comments. Following the reviewer’s suggestion, we performed ThT assay and TEM imaging for tau in HEPES buffer without PEG. The results showed that MB inhibits the conversion of tau droplets into amyloids efficiently in the absence of PEG. The results are shown in Figures 6a and b.

12. Reviewer: *“They have done all the droplet assays without heparin but used heparin in the ThT assay. If heparin is absolutely necessary for mediating tau aggregation, they should show a control for phase separation of tau in the presence of heparin to show that it forms droplets in a manner similar to without heparin.”*

Thanks for the reviewer’s comments. The aggregation of tau is very slow without addition of inducers. Heparin is widely used to accelerate tau aggregation in *in vitro* studies, which facilitates identifying tau aggregation inhibitors, investigating the aggregation mechanism, and testing the toxicity of tau aggregates. Following the reviewer’s suggestion, we investigated the phase separation behavior of tau in the presence of heparin. The results showed that in the presence of heparin the formation of tau droplets is also enhanced with increasing concentrations of MB (Figure S19a and b). We revised the manuscript to discuss the results on Page 15

as follows:

“...We confirmed that MB facilitates tau droplet formation in the presence of heparin (Fig. S19a and b)”

13. Reviewer: “For the TEM images, the 0-hour time point is missing. Were they able to observe any droplets at 0 h in the phase separated conditions?”

Thanks for the reviewer’s comments. We have now added the TEM images of the samples at 0 h in Figure 6b. Under LLPS conditions, the droplets were clearly observed under TEM upon fixing with paraformaldehyde.

14. Reviewer: “Instead of just relying on the ThT assay to make the point about fibrils and doing TEM of the ThT assay samples, which will have contributions from the dilute as well as the dense-phase, they should also try to directly monitor the fibrillization going on in droplets to really make the point that fibrils are not forming in the aging droplets and compare it to a control without MB showing amyloid fibril formation in the droplets.”

Thanks for the reviewer’s comments. According to the reviewer’s suggestion, we directly monitored the fibrillization of tau in the droplets via ThT imaging under LLPS conditions. In the absence of MB, the enhancement of ThT fluorescence was observed in the droplets with increasing incubation time, indicating formation of fibrillar structures within the droplets. In the presence of MB, no ThT fluorescence was observed in the droplets within the detection time. These results demonstrate that tau fibrils form inside the droplets, which is inhibited by the presence of MB. The new data are shown as Figure 6c. We revised the manuscript to discuss the results on Page 15 as follows:

“...Furthermore, we performed ThT fluorescence imaging of the phase-separated tau by a confocal microscopy and found that MB inhibits the formation of ThT-stainable tau aggregates inside tau droplets (Fig. 6c).”

Response to Reviewer #2

1. Reviewer: “The authors conducted NMR experiments to show that MB interacts with several regions in tau. However, the experiment was done in phosphate buffer and not in droplet forming condition. In order to understand the differences in the interaction that inhibits Tau aggregation and promotes Tau LLPS, the NMR experiment should be done in LLPS buffer and compare with the results obtained in phosphate buffer.”

Thanks for the reviewer’s comments. As mentioned in the response to Reviewer#1, we redid the NMR experiments in LLPS-promoting buffers (50 mM Tris at pH 7.4, with and without 5% PEG8000) to make it consistent with other experiments in this study. However, we decided to omit the previous NMR results that were carried out in phosphate buffer because tau LLPS is highly influenced by the ionic strength and solution conditions, therefore the results in phosphate buffer are difficult to compare with the results in the LLPS-promoting Tris buffer. For a

more prudent comparison between MB binding to tau in “strong” versus “weak” LLPS conditions, we performed the tau-MB NMR titrations with and without PEG8000. The new data are shown in Figure 4 and S14, and discussed on Pages 10-11 as follows:

“¹H-¹⁵N SOFAST HMQC spectra⁵⁷ of ¹⁵N-labeled WT tau protein were recorded in the absence and presence of MB (Fig. 4d and S14b). The NMR spectra were recorded at 5 °C to minimize loss of ¹H signal due to solvent exchange⁵⁸. Under these conditions, tau did not form droplets in the absence of MB (Fig. 4e). Addition of MB resulted in the formation of a biphasic sample consisting of suspended tau droplets in a dispersed phase of tau (Fig. 4e). Quantification of the changes in peak intensities upon MB addition revealed an overall reduction in NMR signal intensities involving all domains of tau (Fig. 4f and S14f). A reduction in NMR signal intensity in relation to protein phase separation might arise from the slower tumbling of the protein in the droplet phase and associated shortened transverse relaxation times⁵⁹. In addition, exchange between an NMR-visible state of the protein (i.e. monomeric tau in the dispersed phase) and an NMR-invisible state (i.e. droplets or aggregates) potentially contributes to the reduction in signal. In the case of protein-ligand binding, NMR line broadening may also occur due to exchange between free and ligand-bound states of the protein in the intermediate exchange regime. To assess the impact of droplet formation without MB binding on the NMR spectrum of tau, we collected the ¹H-¹⁵N SOFAST HMQC spectrum of tau in the presence of 5% PEG8000 (Fig. 4d and S14a). PEG induced tau droplet formation (Fig. 4e) and led to ~40% peak amplitude reduction (Fig. 4f and S14d) as well as chemical shift perturbations (Fig. S14e) across all tau domains, consistent with the formation of a slow-tumbling NMR-invisible state (i.e. droplet phase). Notably, the intensity ratio profile for the tau-PEG mixture does not completely correspond to that of the tau-MB sample (Fig. 4f). The prominent lack of overlay for the entire NTD and CTD regions suggest that these domains are the most affected by MB binding. A patch of the MTBD (residues ~330–350) and some residues in the PRD also appear to bind to MB.”

2. Reviewer: *“The authors investigated several MB derivatives and found azure B was much weaker than MB in inducing turbidity of Tau, but stronger in inducing gelation of tau droplets. This is interesting and worth further investigation. Since turbidity is affected by many factors (size, number, et. al.) of the droplets, it is not the best method to quantify phase separation propensity and should be complemented by another method. Sedimentation assay could be used to determine the critical concentration at which LLPS occurs.”*

Thanks for the reviewer’s comments. We agree with the reviewer that it’s inadequate to analyze phase separation using turbidity only. Following the reviewer’s suggestion, we measured the saturation concentrations of tau under different concentrations of MB, azure A, or azure B using centrifugation. We found that the saturation concentration of tau decreased markedly as the

concentrations of MB or azure A were increased, indicating that MB and azure A enhance the capacity of tau to undergo phase separation. In contrast, the saturation concentration of tau showed marginal changes as the concentration of azure B was increased, indicating the minimal effect of azure B on the capacity of tau to undergo phase separation. The saturation concentration measurements together with the FRAP analysis suggest that azure B primarily enhances dynamic arrest of tau condensates whereas azure A and MB enhance dynamic arrest of tau condensates as well as increase the driving force for tau to undergo phase separation. The results are shown in Figures 1d and 3d. We have revised the manuscript to discuss the results as follows:

On Page 4:

“...To assess the effect of MB on the phase separation propensity of tau, we determined the saturation concentration of tau (C_{sat}) under different MB concentrations using centrifugation⁴⁴. As shown in figure 1d, C_{sat} decreased with MB concentration, suggesting that MB enhances the capacity of tau to undergo phase separation.”

On Page 8:

“...Saturation concentration measurements showed that azure A has similar effects as MB on reducing C_{sat} (Fig. 3d). In contrast, C_{sat} only changed marginally as the concentration of azure B was increased, indicating that azure B enhances dynamic arrest of tau droplets with minimal effect on the capacity of tau to undergo phase separation.”

3. Reviewer: *“According to Ref 51, the IC50 of MB and azure B in inhibiting Tau aggregation is the same. But their effect on Tau phase separation is different, therefore, NMR study of Tau structure in the presence of MB and azure B might be able to tell which interactions are responsible for the inhibition activity and which interactions are responsible for the promoting activity.”*

Thanks for the reviewer’s comments. We agree that the comparison between MB and azure B binding to tau is very interesting and would potentially shed light on why these molecules have different activity in inducing gelation of tau droplets, thus we also performed NMR titrations for tau-azure B under the conditions used for tau-MB titrations. The new data are shown in Figures 4 and S14, and discussed on Page 11 as follows:

“...We also collected the ^1H - ^{15}N SOFAST HMQC spectrum of tau in the presence of azure B (Fig. 4d and S14c). Comparing the intensity ratio profiles for tau-MB and tau-azure B (Fig. 4f), we observed that the profiles are mostly consistent, and that I/I_0 attenuation due to MB is more pronounced than that due to azure B except for short stretches of residues such as those around ~200 in the PRD and ~380–400 in the CTD, suggesting that interaction with these residues may be responsible for the different effects of MB and azure B on inducing tau LLPS and droplet gelation. Addition of PEG8000 together with MB or azure B resulted in severe attenuation of the NMR peak intensities due to enhancement of phase separation (Fig. S14g and h), which prohibited further residue-level analysis.”

4. Reviewer: *“The authors showed that a Tau liquid-to-gel transition inhibitor TMAO, at a concentration up to 0.2 M, could not mitigate the liquid-to-gel transition of tau droplets in the presence of MB, azure A or azure B. Some mechanistic insight of this result would be appreciated. Does TMAO engage the same regions as MB? Are there competitive binding?”*

Thanks for the reviewer’s comments. Docking simulations were performed to obtain more details on the interactions between the ligands and tau. The docking results showed that the ligands could bind to various regions of tau protein and tau residues involved in ligand binding overlap among different ligands. The binding energies are -348.78 , -367.20 , -355.78 , and -193.26 kcal/mol for MB, azure A, azure B, and TMAO, respectively, indicating that the interactions between TMAO and tau are much weaker than those for MB, azure A, and azure B. The results are shown in Figures 4g,h, S15, and S16, and discussed on Page 11 as follows:

“To obtain atomic models for the tau-MB complex state, docking simulations were performed. Considering that MB may have a few dominant binding modes with tau, we focused on the top 5 docking poses (Fig. 4g) and calculated the contribution of each residue to ligand binding through docking score decomposition analysis. Consistent with NMR characterization and turbidity analysis, docking results showed that MB interacts with various regions of tau, including the two termini of the NTD, the central region of the PRD, the R2 region of the MTBD, and the N-terminal half of the CTD (Fig. 4h). We further analyzed binding of azure A, azure B, and TMAO with tau using docking simulations (Fig. S15 and S16). Similar to MB, the interactions between azure A/azure B and tau were distributed throughout the tau protein. The total binding energies calculated from all 1000 docking poses were -348.78 , -367.20 , -355.78 , and -193.26 kcal/mol for MB, azure A, azure B, and TMAO, respectively, indicating that the interactions between TMAO and tau are much weaker than those for MB, azure A, and azure B.”

5. Reviewer: *“From the FRET experiments, the authors concluded that conformational expansion of tau in the droplets results from phase separation rather than from MB binding. Based on the data presented, it is clear that binding of MB in non-LLPS buffer does not change the conformation of Tau. However, since interaction between MB and Tau in non-droplet forming condition, is different than in droplet forming condition, we can’t rule out the possibility that in droplets, binding of MB enhances the conformational expansion.”*

Thanks for the reviewer’s comments. We further detected the effect of MB on AF350/AF488-labeled tau under non-phase-separation conditions and phase-separation conditions. The normalized fluorescence spectra of 200 nM AF350/AF488-labeled tau (without unlabeled tau) under non-phase-separation conditions are essentially identical for Tris buffer and HEPES buffer (Figure S18), suggesting that the interactions between MB and tau are the same in the two buffers. The slight decrease in the FRET signals when the concentration of MB

was increased to 100 μ M suggests that MB binding perturbs the conformations of tau. When LLPS occurred, a larger decrease in the apparent FRET signal was observed under 100 μ M MB (a comparison of the red lines in Figures 5b,e and S18), suggesting that phase separation further induces expansion of tau conformation. In order to directly detect the MB effect on the phase-separated tau, we carried out FRET imaging of AF350/AF488 dual-labeled tau in the LLPS state using a two-photon laser scanning confocal microscope with 700 nm laser as the excitation light source for AF350. The AF350 and AF488 fluorescence from the tau droplets was detected simultaneously in two separated channels (Figure 5c). The results showed that upon addition of MB, the AF488/AF350 intensity ratio in the droplets decreases (Figure 5d), suggesting that MB enhances the conformational expansion of tau in the droplets. We have revised the manuscript on Page 13 as follows:

“...In Tris buffer, we observed significant FRET signals for all four tau variants in their native state (Fig. 5b, black lines), indicating that AF350 and AF488 were within the distance that can undergo FRET. This is consistent with the paper-clip model of the native tau conformation^{61,62}. Incubation of tau with MB in the non-LLPS state resulted in slight changes in the apparent FRET signals (Fig. 5b, green lines, and Fig. S18a), suggesting that MB binding perturbed the conformations of tau. This is consistent with the above results from NMR and docking, which indicate extensive interactions between MB and tau (Fig. 4). Inducing tau phase separation by adding PEG decreased the FRET signal of the labeled tau variants (Fig. 5b, blue lines). A further decrease in the apparent FRET signal was observed when tau was co-incubated with MB and PEG, which led to an obvious LLPS state (Fig. 5b, red lines). Two-photon microscopy images further demonstrated that the FRET decrease occurs within the droplets upon addition of MB (Fig. 5c and d). In HEPES buffer, a similar phenomenon was observed although the crowding reagent PEG8000 was not necessary for LLPS. The FRET signals of dual-color labeled tau variants showed a slight decrease in the presence of MB in the non-LLPS state (Fig. 5c, blue lines, and S18b). A marked decrease in the FRET signal was observed when LLPS occurred (Fig. 5c, red lines). To check whether the decrease of the apparent FRET efficiency was caused by artificial quenching of the acceptor (i.e. AF488), we performed fluorescence lifetime measurements, which showed no significant change of AF488 lifetime upon addition of MB in either the native or LLPS states (Table S2). Therefore, our FRET analysis indicates that the distance between the labeling pair increases upon MB-induced LLPS especially for the NTD and CTD regions, indicating that the tau conformation changes to an expanded state, which could result from a synergistic effect of phase separation and MB binding.”

6. Reviewer: *“For figure 7, the authors concluded that gelation of tau droplets induced by MB slightly retards the assembly kinetics of tubulin within tau droplets without affecting the final appearance of microtubule bundles. The conclusion is drawn by visual observation of the fluorescence images. I would appreciate that*

the authors to quantify the assembly kinetics and test the significance of the differences with/without MB.”

Thanks for the reviewer’s comments. To quantify the influence of MB on the microtubule assembly kinetics, the assembly process was monitored by diamidino-phenylindole (DAPI) fluorescence. We confirmed that DAPI has minimal effect on the FRAP recovery of tau droplets. The DAPI fluorescence increased rapidly in the first 10 min and reached a plateau in the absence or presence of MB, indicating the formation of microtubules under both experimental conditions. The half-time to reach the plateau increased from 5.6 min to 7.9 min when MB was added, suggesting that MB slightly retards the assembly kinetics of tubulin. The results are shown in Figures 7d,e,f, and discussed on Page 17 as follows:

“...To quantify the influence of MB on the microtubule assembly kinetics, the assembly process was monitored by the fluorescence of diamidino-phenylindole (DAPI)^{63,64} which has minimal effect on the FRAP recovery of tau droplets (Fig. 7d and e). As shown in figure 7f, the DAPI fluorescence increased rapidly in the first 10 min and reached a plateau phase. The half-time at which the DAPI fluorescence reached 50% of the plateau value increased from 5.6 min to 7.9 min when MB was added (Fig. 7f, inset).”

REVIEWERS' COMMENTS

Reviewer #1 (Remarks to the Author):

The manuscript by Huang et al. explores the question how a small molecule inhibitor of tau fibrillization influences phase separation. The authors use phase separation and fibrillization assays, and also use alternative biophysical approaches to characterize the interaction of the small molecule with tau. They conclude that loss of dynamics of the phase separated state is not necessarily coupled to enhanced fibrillization. This is an interesting result that is important to report in my opinion given the unfortunate convolution of "gelation" and "fibrillization" in the phase separation literature. The authors have added a lot of additional data to address my comments, and I think that the revised manuscript is much improved. Some interpretations are still too strong and I request textual changes to make sure that all data is appropriately interpreted.

1. The authors report an interesting competition between fibrillization and dynamical arrest. It would be of interest to see what happens in the droplets at intermediate concentrations of MB, where ThT assay shows suppression of fibril formation. Is there a combination of fibrillization and dynamical arrest going on? If they have such droplet images, it would be useful and informative to include them in the SI.

2. To quantify the influence of MB on the microtubule assembly kinetics, the assembly process was monitored by the fluorescence of diamidino-phenylindole (DAPI). Although they have cited references, it should be explained for the clarity of the readers in a few sentences what the rationale is behind using DAPI. It would also be useful to show some fluorescence microscopy images demonstrating that tubulin and DAPI colocalize in the droplets.

3. There is a lack of clarity in the manuscript at some places regarding the conditions that have been used for the experiments, which can be confusing for the readers. For e.g. in Supplementary table 2 it is not clear if the fluorescence lifetime measurements have been obtained in phase separated/non phase separated conditions.

Reviewer #2 (Remarks to the Author):

In this revised manuscript, the authors provided a large amount of additional data from NMR experiments, centrifugation assay, and FRET measurement. The authors have addressed all my concerns in this revised manuscript.

A few minor comments:

1. Line 375 and 376: Figure 5C should be Figure 5e.
2. For the quantification of the FRET image, Figure 5d, does individual data point represent each droplet or each image? I would appreciate the information on N of the experiments.

Response to Reviewer #1

1. Reviewer: *“The authors report an interesting competition between fibrillization and dynamical arrest. It would be of interest to see what happens in the droplets at intermediate concentrations of MB, where ThT assay shows suppression of fibril formation. Is there a combination of fibrillization and dynamical arrest going on? If they have such droplet images, it would be useful and informative to include them in the SI.”*

Thanks for the reviewer’s comments. We performed ThT fluorescence imaging for tau droplets supplemented with 1 μ M MB and 10 μ M MB. The results are shown in Supplementary Figure 20. As shown in Figure 6c and Supplementary Figure 20, the intensity of ThT fluorescence inside the droplets decreased as the concentration of MB increased. At 10 μ M MB, ThT fluorescence was observed, although FRAP experiments and aspect ratio measurements (Figure 2) indicated that the tau droplets were in a gel-like state, suggesting the co-occurrence of fibrillization and dynamical arrest.

2. Reviewer: *“To quantify the influence of MB on the microtubule assembly kinetics, the assembly process was monitored by the fluorescence of diamidino-phenylindole (DAPI). Although they have cited references, it should be explained for the clarity of the readers in a few sentences what the rationale is behind using DAPI. It would also be useful to show some fluorescence microscopy images demonstrating that tubulin and DAPI colocalize in the droplets.”*

Thanks for the reviewer’s comments. Following the reviewer’s suggestion, we have explained the usage of DAPI on Page 11 as follows:

“To quantify the influence of MB on the microtubule assembly kinetics, the assembly process was monitored by the fluorescence of diamidino-phenylindole (DAPI)^{63,64}, which binds to tubulin and exhibits fluorescence enhancement upon association. The affinity of DAPI for tubulin increases when tubulin assembles into microtubules, thus leading to a further enhancement in DAPI fluorescence.”

We further demonstrated the co-localization of DAPI and tubulin in the tau droplets using fluorescence microscopy imaging. The results are shown in Figure 7f.

3. Reviewer: *“There is a lack of clarity in the manuscript at some places regarding the conditions that have been used for the experiments, which can be confusing for the readers. For e.g. in Supplementary table 2 it is not clear if the fluorescence lifetime measurements have been obtained in phase separated/non phase separated conditions.”*

Thanks for the reviewer’s comments. We have revised Supplementary Table 2 by adding a column to indicate the sample state.

Response to Reviewer #2

1. Reviewer: “Line 375 and 376: Figure 5C should be Figure 5e.”

Thank you to the reviewer. The error has been corrected.

2. Reviewer: “For the quantification of the FRET image, Figure 5d, does individual data point represent each droplet or each image? I would appreciate the information on N of the experiments.”

Thanks for the reviewer’s comments. The individual data points in Figure 5d represent each droplet. A total of 58 droplets were analyzed. To clarify this, we have revised the legend of Figure 5 as follows:

“...d Apparent FRET efficiency of tau droplets measured by the two-photon microscopy images under excitation of AF350 by 700 nm laser. Individual data points represent each droplet. A total of 58 droplets were analyzed. The bars indicate minimum and maximum. The center and bounds of box indicate median and SD, respectively. Significance levels were determined by unpaired two-sided Student’s t-test. ...”